



# Hotspots for warm and dry summers in eastern Europe, with a focus on Romania

Viorica Nagavciuc[1,2], Patrick Scholz[1] and Monica Ionita[1,3*]
[1]Alfred Wegener Institute Helmholtz Center for Polar and Marine Research, Paleoclimate Dynamics Group, Bremerhaven,
Germany
[2]Faculty of Forestry, Ștefan cel Mare University, Suceava, Romania
[3]Emil Racovita Institute of Speleology, Romanian Academy, Cluj-Napoca, Romania
**\* Correspondence:**
Monica Ionita
Monica.Ionita@awi.de
**Keywords: heatwaves, drought, compound events, atmospheric circulation, climate change.**
**Abstract**
The combined effect of hot and dry extremes can have disastrous consequences for the society, economy, and the environment.
While a significant number of studies have been conducted regarding the variability of the individual hot or dry extremes in
Romania, the evaluation of the combined effect of these extremes (e.g. compound effect) is still lacking for this region. Thus,
in this study we have assessed the spatio-temporal variability and trends of hot and dry summers in the eastern part of Europe,
focusing on Romania, between 1950 and 2020and we have analyzed the relationship between the frequency of hot summers
and the prevailing large-scale atmospheric circulation. The length, spatial extent and frequency of HWs in Romania has
increased significantly over the last 70 years, while for the drought conditions no significant changes have been observed.
The rate of increase in the frequency and spatial extent of HWs has accelerated significantly after the 1990's, while the
smallest number of HWs was observed between 1970 and 1985. The hottest years, in terms of heatwave duration and
frequency, were 2007, 2012, 2015, and 2019. One of the key drivers of hot summers, over our analyzed region, is the
prevailing large-scale circulation, featuring an anticyclonic circulation over the central and eastern parts of Europe and
enhanced atmospheric blocking activity associated with positive temperature anomalies underneath. We conclude that our
study can help improve our understanding of the spatio-temporal variability of hot and dry summers, especially at the regional
scale, as well as their driving mechanisms which might lead to a better predictability of these extreme events.



## 1 Introduction

According to the recently published AR6 report (IPCC, 2021): "It is virtually certain that there has been increases in the intensity and duration of heatwaves and the number of heatwave days at the global scale". This tendency has been clearly observed, especially over the last two decades, when a significant increase in the frequency of hot summers has been observed (Feng et al., 2020; Raymond et al., 2020; Seneviratne et al., 2012; Zscheischler et al., 2018). In some regions, these hot summers were accompanied by extremely dry conditions, leading to the development of the so-called "compound events" (Leonard et al., 2014). Overall, heatwaves and droughts fall into the category of climate related hazards which affect more and more frequently socio-economic activity, often having serious repercussions on humans and the environment (IPCC, 2021). Thus, in the context of the ongoing climate change, the study of heatwaves and droughts and the analysis of the large-scale circulation patterns which favors their occurrence is of increasing interest (Feng et al., 2020; Geirinhas et al., 2021; Ionita et al., 2021a; Kong et al., 2020; Russo et al., 2019).

Several studies have suggested that due to global warming the large-scale atmospheric circulation has been altered both regionally and globally (Horton et al., 2015; Vaideanu et al., 2020). Any perturbation in the large-scale atmospheric circulation will also lead to changes in the hydroclimate, due to the fact that the atmospheric circulation plays a crucial role in the global and regional hydroclimatic variability (Ionita et al., 2020; Kingston et al., 2006, 2015; Schubert et al., 2016). Changes in temperature and precipitation have been found to be a direct response to changes in the large-scale atmospheric circulation patterns (e.g. an increase in the frequency of blocking conditions or an intensification of the westerlies) (Horton et al., 2015; Rimbu et al., 2014; Swain et al., 2016). For example, one key driver of the European hydroclimate variability is the prevalence of long-lasting high-pressure systems (also known as atmospheric blocking) (Bakke et al., 2020; Barriopedro et al., 2011; Ionita et al., 2021b; Kautz et al., 2021; Rimbu et al., 2014; Schubert et al., 2014). These long-lasting high-pressure systems have a significant impact on different types of extreme events such as heatwaves (Barriopedro et al., 2011; Della-Marta et al., 2007; Laaha et al., 2017), cold spells (Jeong et al., 2021; Rimbu et al., 2014), droughts (Ionita et al., 2012; Kingston et al., 2015; Schubert et al., 2016) and floods (Grams et al., 2014; Najibi et al., 2019). Thus it is essential to study the relationship between the changes in the magnitude and frequency of extreme events and their large-scale drivers, in order to have a better overview of the physical mechanisms leading to the occurrence of these extreme events.

In terms of exposure and vulnerability to such climate-related risks (e.g. heatwaves and droughts), Romania is particularly prone, both due to its geographical position, as well as the topographic features, which give it a very special status in relation to the manifestations of the weather (Croitoru and Piticar, 2013; Micu et al., 2021; Sfîcă et al., 2017). The existence of the Black Sea and, especially, the concentric distribution (i.e. "in the amphitheater") of the Carpathian Mountains (Figure 1), induce a series of peculiarities in the prevailing climatic conditions that are also reflected in the thermal regime mediated at the scale of different regions of the country. Moreover, the evolution of the weather in Romania depends strongly on the degree of exposure to alternating, often rapid, types of air masses passing the country (e.g. continental, tropical, maritime, or polar) (Bădăluță et al., 2019; Busuioc et al., 2010, 2015; Tomozeiu et al., 2005).

At country scale, different studies have analyzed the potential changes in the frequency of HWs, either by using observational records (e.g. station data) or gridded datasets (Croitoru et al., 2016b; Croitoru and Piticar, 2013; Hustiu, 2016; Micu et al., 2021; Sfîcă et al., 2017). In their paper, Sfîcă et al. (2017) have analyzed the synoptic conditions which lead to the occurrence



of heatwaves in Romania, over the period 1961 – 2015. By analyzing 111 HW events they found that there are two major
types of weather patterns associated with HW occurrence, namely positive or neutral sea level pressure anomalies and
persistent ridges, over the analyzed region. Over the same period (i.e. 1961 – 2015), Croitoru et al. (2016) found that the
frequency of heatwaves, defined based on the daily maximum temperature, shows a significant increasing trend, especially in
the western and central part of Romania. Looking at a more regional scale, Croitoru and Piticar (2013) have shown that there
is an increasing trend in the frequency of heatwave events over the extra-Carpathian regions of Romania (i.e. the eastern and
southern part of the country) and that the daily maximum temperature is getting more extreme compared to the daily minimum
temperature. Over the eastern part of the country, Hustiu (2016) has shown that the annual frequency of heatwave events
features an increasing trend over the period 1961 – 2013, while in a more recent study, Micu et al. (2021) have shown that
the southern part of the Carpathian Mountains is facing a significant warming trend. All the aforementioned studies are either
limited in time or are very regional (Croitoru and Piticar, 2013; Hustiu, 2016; Sfîcă et al., 2017; Spinoni et al., 2015) and they
were mainly focused on the analysis of trends in the heatwave frequency. To our knowledge no in-depth analysis, for this
region, has been made regarding the variability and trend for compound events (e.g. hot and dry summers). Moreover, taking
into account that the frequency of extreme events (e.g. heatwaves, cold spells, drought, and floods) is projected to increase in
the future (IPCC, 2021) it is imperative to understand the physical process forcing the increase in the frequency and magnitude
of these events in order to improve their predictability. Therefore, in this study we aim to make an in-depth analysis of the
spatio-temporal variability of both hot and dry summers over Romania and to analyze the relationship between the frequency
of hot summers and the prevailing large-scale atmospheric circulation. The paper is focused on two main objectives: i) to
analyze the trends and the spatio-temporal variability of both hot and dry summers in Romania, as well as their combined
effect (e.g. compound events) and ii) to determined the large-scale circulation patterns which trigger the occurrence of hot
summers over the analyzed region, by analyzing the geopotential height conditions and the frequency of atmospheric blocking
during the periods characterized by a high frequency of hot days.  Our study extends over the period 1951 – 2020, making it
the most extensive study, from a temporal point of view, over eastern Europe. The paper is structured as follows in Section 2
we give a detailed description of the data and methods used in this study. In Section 3 we show the main results of our analysis,
while the main conclusions are presented in Section 4.
**2 Data and methods**
Globally, heatwaves are recognized either by utilizing a threshold-based methodology (Perkins and Alexander, 2013) or by
using the exceedance of a fixed absolute value (e.g. daily maximum temperature > 30°C) (Robinson, 2001). In general, the
method based on fixed thresholds takes into account periods of consecutive days when the daily maximum temperature (Tx)
is above a certain percentile for a particular calendar day. In this study, we have used the 90th percentile, based on a 15-day
window centered on each calendar day (Perkins and Alexander, 2013). For the duration, we have tested different lengths of
3, 4, 5, and 6 consecutive days (Figure S1).  The mean daily $90^{th}$ percentile was calculated over the baseline period 1971 –
2000. The daily maximum temperature used in this study was extracted from the E-OBSv23.1e data set (Cornes et al., 2018).
Here, the heatwave duration index (HWDI) is defined as the number of days per month/season when the afore-mentioned
criteria were satisfied, while the number of heat waves (HW) is defined as the number of heatwaves per month/season. The
temporal evolution of the HWDI for each summer month (i.e. June, July, and August) as well as for the whole summer season



(JJA), for all considered lengths (Figure S1), indicate a strong interannual variability and relatively significant decadal
differences. As expected, the smaller the length of the threshold, the longer the heatwave. Globally, different duration
thresholds have been employed, depending on the analyzed regions. For example, in Canada, a duration threshold ≥2 days
has been used (Smoyer-Tomic et al., 2003), in Hungary and France a duration threshold ≥3 days has been considered (Rey et
al., 2007), while in China and Ukraine a duration threshold ≥ 5days has been used (Chen and Li, 2017; Shevchenko et al.,
2014). In the eastern part of Europe (e.g. Bulgaria), a duration threshold ≥3 days has been found useful (Gocheva et al., 2006).
Since Romania is situated in the eastern part of Europe, where a threshold ≥ 5 or 6 days has been tested and because we want
to analyze extreme heatwave in this study, the rest of the analysis is focused on a threshold ≥5 days.
The hydroclimatic conditions, with a special focus on the drouth component (values < -1), are defined by considering the 1-
month and 3-month Standardized Precipitation Index (SPI3) (McKee et al., 1993). For this analysis, we used the June, July,
and August SPI1 index and the August SPI3 index, which integrates the drought conditions over the whole summer months
(i.e. June-July-August). The SPI index was extracted from the E-OBS v23.0e data set, with a spatial resolution of 0.1° x 0.1°
and a temporal resolution covering the period 1950 – 2020. SPI is based on the accumulated precipitation data, where the
precipitation (PP) data is fitted to a gamma distribution (McKee et al., 1993). The advantage of using SPI is that it is
standardized on a given period and a predefined distribution. Therefore, each SPI value corresponds to a predefined
probability. Here, we choose the threshold of -1 meaning that all occurrences of SPI below the threshold would be considered
as drought. This threshold generally corresponds to a moderate to extreme drought event. Taking into account our definition
HW and drought, a compound hot and dry (CHD) event is defined as a combined index when a heat wave episode occurs
during a period of drought conditions (i.e. a summer with an associated 3 month SPI value (SPI3 August) <-1). This definition
has also been used successfully for other regions (Geirinhas et al., 2021; Ionita et al., 2021a; Russo et al., 2019). For our
analysis, we only use the grid points that meet the criteria for compound events.
To compute the SPI, HWDI, and CHD we make use also of a regional dataset covering Romania, namely the ROCADA data
set (Dumitrescu and Birsan, 2015). ROCADA is a daily gridded observational data set for minimum, mean and maximum
temperature, precipitation, soil temperature, sea level pressure, relative humidity, cloud cover, and sunshine duration, covering
Romania, based on station information. The data set covers the period 1961–2013 and the spatial resolution of the dataset is
0.1° x 0.1°. Because the ROCADA data sets have a lower temporal extent compared to the EOBS data set, most of the analyses
in this study are based on the EOBS dataset. A detailed comparison between the EOBS and ROCADA dataset is given
throughout the manuscript.
To analyze the large-scale driving mechanism of heatwaves, we use the daily temperature at 850mb level (TT850), the daily
geopotential height at 500mb level (Z500), as well as the daily zonal and meridional wind at 500mb level. These datasets have
been extracted from the ERA5 reanalysis project (Hersbach et al., 2020), and have a spatial resolution of 0.25◦ × 0.25◦,
covering the 1950–2020 period.



## 3 Results

### 3.1 Summer heat waves in eastern Europe: variability and trends

The heatwave duration index (HWDI) averaged at the country level and the fraction of the country affected by the heatwave (AREA) are shown in Figure 2. This figure reveals that there is strong interannual and decadal variability throughout all summer months (Figure 2- left column). For June, there is a statistically significant increase in both HWDI (Figure 2a and Table S1) and AREA (Figure 2b), which is a much higher frequency of both after the beginning of the 1990's. The longest heatwave was recorded in June 2019 and lasted 10 days, when more than 90% of the country was affected. Until the beginning of the 1990's there were relatively few HWs, most o them observed between 1960 and 1970, but their duration is much smaller compared to the events recorded from 2000 onwards. Also in terms of the affected area, after 1990's most of the heatwaves had a larger spatial extent, with an area covered by a HW of more than 80% in 1996, 2002, 2003, 2010, 2012, and 2019, respectively.

As in the case of June, for July we observe also a statistically significant increase both in the HWDI (Figure 2c and Table S1) and the AREA (Figure 2d). At the beginning of the analyzed period (i.e. 1950 – 1960), there were heatwave events lasting on average 4 – 5 days (when averaged at country level) and covering an area up to 80%. Between 1970 and 1985 no HWs were recorded throughout the country. After 1985 there is a steep increase in the duration of the HWs, with the longest HWs in July 2007 and 2012, when the whole country was affected (i.e. AREA = 100%). Years 1987, 2002, 2007, 2012, and 2015 have been characterized by HWs with a spatial coverage of more than 80% (Figure 2d). For August, the temporal evolution of HWDI (Figure 2e) and AREA (Figure 2f) follows the same path as June and July, meaning a significantly increasing trend in both the duration (Table S1) and area (not shown) after 1995. Over the period 1964 – 1988 no HWs has been recorded in August, while most of the longest and extended HWs were recorded in the last two decades of the analyzed period. The longest HW recorded in August was in 2015, followed by 2012, 1992, and 1952. In 1992, 2012, and 2015 the area covered by HWs was higher than 90% (Figure 2f). For all analyzed months, the HWs recorded in the last two centuries were both longer and had a higher spatial extent. If we analyzed the whole summer months have taken together (JJA), we have a very clear picture (Figure 2g and 2h): the period 1950 – 1970 was characterized by the occurrence of HWs with an average duration of 10 days and a spatial extent between 20% up to 80%, followed by a relatively HW free period between 1970 and 1985. After this period there was a significant increase in the duration of the HWs and most of them reached a spatial extent of more than 50%, especially over the last decades (i.e. 2001 - 2020). In terms of used data sets, both EOBS and ROCADA datasets capture in a similar manner the temporal distribution of the HWDI and AREA. The correlation coefficients between the monthly HWDI (AREA), computed based on the two datasets (i.e. E-OBS and ROCADA), is >0.9 for all months, thus the EOBS dataset is able to fully capture the spatio-temporal variability of the HWs, allowing us to extend our analysis for 70 years. This finding is supported also by other studies which have used the EOBS data set for regional analysis over Romania (e.g. Ionita et al., 2015; Sidău et al., 2021). For example in their study, Sidău et al. (2021) have shown that the EOBS data set captures the best, when compared to observations, the local variability, in terms of temperature and precipitation, over Romania.

Since the number of HWs per year is small, especially in the first analyzed period (i.e. 1951 – 1985), we have aggregated the number of heatwaves in decades, to be able to analyze the spatio-temporal changes in their occurrence. We have performed



the decadal analysis for each summer month separately (Figure S2, S3, and S4) and for the whole summer season (JJA) as a
whole (Figure 3). We focused our analysis in his way, to have an equal number of months/decade and also to provide decadal
evolution of HWs hotspot, at country level. The first analyzed decade is 1951 – 1960, followed by 1961 – 1970, and so on
until 2011 – 2020. Figure 3 shows that the geographical distribution of the number of HWs/decade summed over the summer
months. Overall, there is an increased variability among different regions of the country depending on the analyzed decades.
Over the decade 1951 – 1960 up to 24 HWs/decade have been recorded in the south-eastern part of the country (i.e. the
Dobrogea region), while in the north-west part of the country up to 10 HWs/decade have been recorded (Figure 3a). Over the
decade 1961 – 1970 HWs (up to 8 HWs/decade) have been recorded mainly in the Intra-Carpathian region (i.e. the north-west
part of the country) (Figure 3b). The decade 1971 – 1980 was almost HWs free (Figure 3c), while for the decade 1981 – 1990
there were less than 2 HWs/decade at country level (Figure3d). Starting with the decade 1991 – 2000 the number of summer
HWs started to increase all over the country (Figure 3e). During the 2001 – 2010 decade, the HW hotspots developed in the
western part of the country and the Doborgea region (i.e. south-eastern part of the country) (Figure 3f). Over the decade 2011
– 2020, there were up to 24 HWs/decade, the most affected areas being the north-western part, inside the Carpathian Chain,
and the south-eastern part of the country (Figure 3g). Overall, there was up to 6 times more HWs in the last decade compared
to the HWs at the beginning of the analyzed period.
When looking at the decadal distribution of HWs hotspots for each summer month separately, there are some clear differences,
especially at the beginning of the analyzed period (Figure S2, S3, and S4). Over the decade 1951 – 1960, there were up to 5
HWs/decade in July (Figure S3a) and August (Figure S4a), focused in the north-western part of the country and the most
south-eastern corner of the country. In June, a limited number of HWs have been recorded in this decade (~2 HWs/decade)
over the eastern part of the country (Figure s2a). The decade 1961 – 1970 was characterized by up to 4 HWs/decade in June,
over the north and north-western part of the country (Figure S2b), while in July (Figure S3b) and August (Figure S4b) 1
HW/decade was recorded in the western part of the country. The decade 1971 – 1980 was HW free in July (Figure 3c) and
August (Figure 4c), while in June there were ~1 HW/decade over a small part of the country (Figure S2c). The decade 1981
– 1990 was characterized by up to 2HWs/decade, at country level, in July (Figure S3d) and August (Figure S4d). Starting
with the 1991 – 2000 decade, the number of HWs/decade starts to increase at the country level, the most affected months
being June (Figure S2e - 2g) and August (Figure S4e - 4g). Over the decade 2001 – 2010 there were up to 7 HWs/decade
recorded in the south and south-eastern part of the country in June (Figure S2f), up to 6 HWs/decade in the western part of
the country and the Dobrogea region, in July (Figure S3f) and up to 10 HWs/decade in August, with a focus on the Dobrogea
region (Figure S4f). For the last decade (i.e. 2011 – 2020) the number of HWs/decade has increased in all months, but their
spatial distribution differs. In June (Figure S2g), the highest number of HWs/decade was recorded over the north-western part
of the country (up to 10 HWs/decade) and in Doborgea region. In July, the HWs hotspots are over the northern and eastern
part of the country (Figure S3g), while in August there is a homogenous distribution of up to 10HWs/decade, throughout the
country (Figure S4g).
From the decadal analysis of the number of HWs, we can clearly state that the decade 2011 – 2020 was characterized by a
significant increase in the number of HWs compared to the previous decade, this increase being the strongest in August. There
are preferred hotspots for the HWs occurrence, depending on the analyzed decade and month, these hotspots being strongly
influenced by the geographical distribution of the Carpathian Mountains. The most affected regions by the HW occurrence



are the north and north-western part of the country and the Dobrogea region. Dobrogea region is a region which has been
subjected to a significant increase in the mean air temperature and a significant decrease in the summer precipitation (Chelcea
et al., 2015; Prăvălie et al., 2017). Overall, there is a significant increase, of ~0.2HWs/decade in June, over most parts of the
country, except some small regions in the north-eastern part (Figure 4a). In July a significant increase of ~0.1 HWs/decade
can be observed in the northern part of the country, while for the rest of the country no significant changes have been recorded
(Figure 4b). In August, there is a significant increase in the number of HWs over the analyzed region, especially over the
eastern part of the country (~0.2 HWs/decade). When we consider all summer months together, the increase in the number of
HWs is significant at the country level, with an increase of up to 0.4HWs/decade in the eastern part of the country.

**3.2 Summer droughts in eastern Europe: variability and trends**

Since the aim of this study is to analyze both warm and dry summer in the eastern part of Europe, in this section we evaluate
the changes in the occurrence of drought conditions. The drought conditions are analyzed by employing the standardized
precipitation index with a 1-month accumulation period to represent the drought conditions at monthly scale (i.e. SPI1 June,
SPI1 July, and SPI1 August) and with a 3-month accumulation period to analyze the drought conditions over the whole
summer (i.e. SPI3 August). To analyze the variability and trends of drought conditions, at the country level, we performed
the same analysis like in the previous section: we averaged the SPI at the country level (Figure 5), we performed also the
decadal analysis (Figure 6), and the trend analysis (Figure 7). The temporal evolution of June SPI1 (Figure 5a), July SPI1
(Figure 5c), August SPI1 (Figure 5e), and August SPI3 (Figure 5g) indicates a strong interannual variability of drought
conditions, at the country level, and no significant drying or wetting trend. For June, the driest years, both in terms of amplitude
(Figure 5a) and spatial coverage (AREA, Figure 5b) were: 1950, 1968, 2000, and 2002. In 2002 the whole country was
affected by drought, this summer being of the driest summers on record for Romania (Ionita et al., 2016). For July, the driest
years were: 1952, 1956, 1989, and 2015 (Figure 5c), respectively. For these years, the drought conditions extended to more
than 60% of the country (Figure 5d). In August, the driest years, at the country level were recorded in 1952, 2000, 2002, and
2018 (Figure 5e), with the drought conditions covering more than 60% of the country (Figure 5f). August SPI3, which is an
indicator of drought conditions over the whole summer, indicates that the driest summer, over the eastern part of Europe,
were: 1950, 1952, 2000, 2002, and 2018, respectively (Figure 5g). For all these summers, the drought was covering more than
60% of the country (Figure 5h).
The drought hotspots, at a decadal scale (Figure 6), indicate strong spatio-temporal variability between the different analyzed
decades and between different regions of the country. Over the 1951 – 1960 decade (Figure 6a), the drought hotspots (defined
as the number of months/decade when SPI<-1 for each grid point) was focused in the north-eastern part of the country (Figure
6a). For this period, there were up to 6 summers/decade characterized by drought conditions over these regions. For the decade
1961 – 1970 (Figure 6b) and 1971 – 1980 (Figure 6c) there were a relatively limited number of dry summers (~2 dry
summers/decade) throughout the country, mostly focused on the north-western part and south-eastern part. For the decade
1981 – 1990 (Figure 6d), there is a rather homogenous pattern at the country level, with up to 3 dry summers/decade affecting
the whole country. The decade 1991 – 2000 (Figure 6e) indicates a hotspot for drought in the northern part of the country (~6
dry summers/decade), while for the rest of the country there were up to 2 dry summers/decade. The decade 2001 - 2010 is
characterized by a limited number of dry summers, with a focus on the south-eastern part (Figure 6f). Over the 2011 – 2020



decade, the drought hotspots are located mainly over the northern part of the country (Figure 6g). Overall, the decadal spatio-
temporal evolution of the drought conditions (Figure 6) indicates that drought are not homogenous throughout the country
and that the decades with the highest number of dry summers were 1951 – 1960 (over the eastern part of the country) and
1991 – 2000 (over the northern part of the country). A similar inhomogeneous pattern is observed when looking at the SPI
trends, both at monthly (Figure 7a-c) and seasonal time scale (Figure 7d). Overall, in June there is a non-significant wetting
trend over the central and north-eastern part of the country and a non-significant drying trend over the northern and southern
part of the country, and Doborgea region (Figure 7a and Table S2). In July there is an overall wetting trend at the country
level, with small exceptions in the northern most part of the country, but the wetting trend is significant only over small areas
in the eastern part of the country (Figure 7b). In August, the spatial trend pattern is rather distinct compared to June and July,
the northern half and the eastern part of the country being characterized by a drying trend, while for the rest of the country
there is no clear signal (Figure 7c). August SPI3 trend, which takes into account all summer months, follows the features
identified for each month analyzed separately: a drying trend over the northern most part of the country and in the northern
part of Doborgea, and a wetting trend over the rest of the country (Figure 7d). The inhomogeneous spatial distribution for
SPI1 and SPI3 (Figure 6 and 7) indicate that drought conditions in Romania are very divers, from a spatial point of view, over
the analyzed period. This finding is in agreement with previous studies which have shown that there is no spatial consistency
in the occurrence of droughts, based on the SPI, over Romania (Cheval et al., 2014; Ionita et al., 2016) and also at European
level (Ionita and Nagavciuc, 2021; Vicente-Serrano et al., 2021).
**3.3 Historical evolution of compound events (e.g. warm and dry summers) in eastern Europe**
Over different regions of the world, hot summers are usually accompanied by extremely dry conditions, leading to the
development of the  so-called "compound events" (Feng et al., 2020; Geirinhas et al., 2021; Leonard et al., 2014; Ridder et
al., 2020; Russo et al., 2019). These compound events have the tendency to occur at the same time or in sequence, leading to
devastating consequences for the society, economy, and environment (Raymond et al., 2020; Zscheischler and Seneviratne,
2017). In this sub-section, we analyze the spatial distribution and the decadal variability of such compound events (i.e. hot
and dry summers), over the period 1951 – 2020. In a first step, in Figure 8 we have computed the total number of HWs (Figure
8 - left column), the total number of months when the SPI < -1 (indicating dry months) (Figure 8 – middle column), and the
total number of months when we had both hot (e.g. a grid point was characterized by a HW) and dry conditions (e.g. a grid
point was recoding SPI values < -1) (Figure 8 - right column). The analysis was performed over the whole period 1951 – 2020
and for June (Figure 8a - 8c), July (Figure 8d - 8f), August (Figure 8g - 8i), and JJA (Figure 8j - 8e). From a climatological
point of view, in June, HWs have a tendency to occur mostly in the western part of the country, while dry conditions tend to
occur over low altitude regions (Figure 8a). Over the Carpathian Mountains, the number of dry years is much smaller
compared to the region located at low altitudes (Figure 8b). When looking at the combined effect of hot and dry summers
(CHDs) (Figure 8c), we observed that these compound events, in June, tend to occur in the western part of the country
(following the climatology of HWs) as well as in the south and south-eastern part of the country (following the climatology
of droughts). In July, HWs tend to occur mostly in the western part of the country, as well as in Dobrogea region (Figure 8d),
while dry years tend to occur mainly on a longitudinal band, in the central part of the country from south to north (Figure 8e).
The CHDs, in July, follow again the same spatial distribution as the ones of the HWs, with the highest number of CHDs being
recorder in the western and northern part of the country (Figure 8f). In August, HWs tend to occur at country scale (Figure



8g) with the highest number in the south-western part of the country, the Dobrogea region and the eastern part of the country
(~30 HWs/70 years). The drought conditions are distributed  all over the country, with small exception in the northern part,
where the frequency of dry years is much smaller compared to the rest of the country (Figure 8h). The frequency of CHDs, in
August, is mainly focalized in the southern part of the county, with the highest amplitudes (~8 CHDs/70 years) in the south-
western and south-eastern part (Figure 8i). When analyzing the whole summer months together, a very clear pattern emerges
in the case of HWs: the most affected area, thought the whole summer is the western part of the country and the Dobrogea
region (~70 HWs /70 years) (Figure 8j). In summer, drought conditions tend to focus in the northern part of the country and
over smaller areas in the south-eastern part of the country (Figure 8k). As a consequence, most of the CHDs, have occurred
mainly in the western and central part of the country (Figure 8e), over the last 70 years (~12 CHDs / 70 years).
Looking at decadal time scale, and summing over the whole summer months (i.e. June, July, and August) over the period
1951 – 1960, CHD events occurred throughout the whole country with small exceptions in the south-eastern part (Figure 9a).
Over the decade 1961 – 1970 the frequency of CHDs was smaller compared to the previous decade as well as their spatial
extent (Figure 9b). The decade 1971 – 1980 was CHD free (Figure 9c), while over the decade 1981 – 1990 there were up to
3 CHDs over a small region in the southern part of the country and 1 CHD at the country level (Figure 9d). Over the decade
1991 – 2000, the CHD events were more frequent (up to 4 CHDs/decade) especially in the northern part of the country (Figure
9e). Throughout the decade 2001 – 2010 (Figure 9f) at least 1 CHD/decade was recorded over most of the country, with  small
exceptions in the eastern part, where no CHD was recorded. Compared with the previous decades, the period 2011 – 2020
(Figure 9g) is characterized by a more homogenous pattern, with CHDs occurring all over the country. The highest number
of CHDs (~3 CHDs/decade) was recorded in the northern regions (Figure 9g). Because the SPI has a very inhomogeneous
spatial pattern, making it rather difficult to match the regions where also HWs occur, the number of CHDs is relatively small
at the country level, and no clear trend has been observed in their frequency (not shown).
**3.4 Extreme heatwave events and their driving factors**
The analysis of the temporal variability of the HWDI and AREA (Figure 2) has emphasized some extreme HWs for each
analyzed month, both in terms of duration and coverage. Thus, in this sub-section, we make a detailed analysis for the longest
HW for each month, in terms of extremeness (e.g. rank maps) and large-scale driving factors. The analysis is focused on three
distinct cases: July 2012, August 2015, and June 2019, respectively.
July 2012 was marked by persistent heat waves, which have determined extremely high temperatures at the beginning of the
month in the western part of the country, afterwards extending to all regions, but especially in the plain and plateau areas
(Figure 10a). In some regions of the country (e.g. eastern and central part) the duration of the HWs was up to 24 days. In
terms of drought, most of the country was affected by moderate to extreme drought in July 2012 (Figure 10b), with small
exceptions in the western part of the country. July 2012, was the hottest month on record (e.g. over the period 1950 – 2020)
over most of the country (Figure 10c). In July 2012, 114 meteorological stations through the country recorded temperatures
above 35°C (Dima et al., 2016). Over the central part of the country, from the south to the north, July 2012 was both hot and
dry (Figure 10d). The peak of the heatwaves was recorded in the last week of the month (Figure S5). Starting with the 23$^{rd}$ of
July, the atmospheric circulation was characterized by a south-easterly flow, which led to an advection of tropical air masses,
generated over the Arabian Peninsula and extending to Russia (Figure S6). At the country level, this large-scale atmospheric





pattern resulted in the establishment of an excessive thermal regime and an increase in the number of hot days (i.e. daily
temperatures > 35°C), especially in the southern and eastern regions (Figure 10e and S5). Between the 26th and 29th of July
2012, the daily maximum temperature up to 10°C was higher, compared to climatology, especially in the eastern part of the
country (Figure S6d – S6g). These excessive temperatures were driven by the persistence of a high pressure system over the
eastern part of Romania and the presence of an atmospheric blocking center over the western part of Russia (contour lines in
Figure S6).
The heat wave and drought event observed throughout the summer of 2015, affected a large portion of continental Europe
and was one of the most severe dry and hot summers over the observational period (Ionita et al., 2017; Laaha et al., 2017;
Van Lanen et al., 2016). Record high temperatures were observed throughout the whole summer over different parts of Europe.
Extremely high temperatures already started to be recorded in June 2015 over the Iberian Peninsula, central and eastern
France, the western Alps, and Ukraine. The heatwave and drought conditions extended towards the central part of Europe in
July 2015 (Ionita et al., 2017). By August 2015, the heat wave moved and continued to develop in central and eastern Europe,
including Romania. For most of the month of August 2015, Romania was under the influence of extremely high temperatures.
The first heat waves occurred between the 3rd and 16th of June (not shown).  Between the 17th and 23rd of August, a short relief
was observed, with temperatures below the climatological mean (not shown). The second and most intense heat waves (e.g.
in terms of the temperature anomalies) started to develop on the 24th of August reaching it's peak at the end of the month
(Figure S7). The longest heat wave was recorded over the northern and eastern parts of the country (Figure 11a). In some
regions in the eastern part of the country, there were up to 24 days which fulfilled the HW definition. Overall, the drought
conditions in August 2015, were not as intense as in July 2012. Only the northern part of the country experienced both heat
wave and drought at the same time (Figure 11b and 11d). August 2015, was also the hottest month on record (e.g. over the
period 1950 – 2020) in the northern and north-eastern part of the country (Figure 11c). The extremely high temperatures
recorded, especially in the last week of August 2015 were mainly driven by the prevailing large-scale circulation. The two
long-lasting heatwaves in August 2015 were determined by the extension of the North African ridge over most of the European
continent (Figure 11e and Figure S8). During the peak of the second heatwave (i.e. 28.08 – 31.08.2015) the eastern part of
Europe was affected by a persistent atmospheric blocking system (contour lines in Figure S8), which was centered over
Romania. This persistent blocking system led to the advection of hot and dry air from the south. Moreover, the anomalous
$Z500$ center over the eastern part of Europe (Figure 11e and S8) suggests a dominant subsidence and adiabatic warming,
reduced cloudiness, and increased incoming solar radiation, thus leading to excessive temperatures over the affected regions.
For the month of June, the longest and largest (in terms of spatial extent) HW event was recorded in June 2019 (Figure 2e –
2f). According to Copernicus (https://climate.copernicus.eu/surface-air-temperature-june-2019) June 2019 was the hottest
June on record both globally and for  Europe, with the central and eastern Europe particularly warm throughout the whole
month. In June 2019, the north-western and south-eastern parts of Romania were the most affected regions by extreme
temperatures (Figure 12a and 12c). Record breaking temperatures were recorded in the most northern part of the country as
well as in the Dobrogea region (Figure 12c). These record breaking temperatures were corroborated with drought conditions
(Figure 12b and 12d). The eastern, central, and south-western parts of the country were less affected by extreme temperatures
(Figure 12a and 12c) and these regions were characterized by wet conditions throughout the month (Figure 12b). The
particular spatial pattern was mainly influenced by the spatial pattern of the large-scale atmospheric circulation (Figure 12e).



The atmospheric circulation at the peak of the heatwave event (Figure S9 and S10) was characterized by a persistent wave-
like pattern extending from the North Atlantic Ocean towards Eurasia (Figure 12e and S10). Positive (negative) geopotential
anomalies were observed over eastern Europe (central North Atlantic and central Siberia) corresponding to the local positive
(negative) temperature anomalies underneath (Figure 12e and S9). The spatial structure of the Z500 field resembles the classic
omega blocking circulation (Figure S10 - contour lines). This pattern favors the advection of warm air from the Sahel towards
the eastern part of Europe and enhances the incoming solar radiation, leading to extremely high temperature anomalies
underneath the high pressure system.
All analyzed extreme HWs in this section were mainly driven by the presence of a high-pressure system over the analyzed
region, during the peak of the HW event. In order to identify if the presence of a persistent high pressure system is a necessary
ingredient for all HWs identified throughout the period 1950 – 2020, we have computed the composite maps (See
Supplementary file for the composite maps definition) for the years when the HWDI index (Figure 2 – left column) was >5
days and the corresponding Z500 anomalies and the corresponding wind vectors. We performed the analysis for each month
separately (Figure S11). Due to the fact that the relationship between the large-scale atmospheric circulation and the European
hydroclimate was found to be limited due to non-stationarity issues (Ionita et al., 2020; Rimbu et al., 2004; Vicente-Serrano
and López-Moreno, 2008), we have computed also the stability map s between the HWDI and the monthly Z500. The aim of
the composite map analysis is to analyze the relationship between the HWDI and the large-scale atmospheric patterns, but
this methodology does not consider if the relationship between the two variables is stationary in time or not. In order to
overcome the problem of non-stationarity and to test if the identified relationship between the HDWI and Z500 is stable over
time, we employed a methodology, namely the stability maps, used for the monthly to seasonal prediction of the mean runoff
of the Elbe River and in dendroclimatological studies (Ionita et al., 2015a; Nagavciuc et al., 2019). A detailed description of
this methodology is given in the aforementioned papers.
The June composite map of Z500 anomalies and the corresponding wind vectors for years with HWs lasting more than 5 days,
is characterized by positive Z500 anomalies over the central and eastern part of Europe and negative Z500 anomalies over the
central North Atlantic Ocean (Figure S11a). Moreover, HWs in June, in Romania, are also associated with an increase in the
number of atmospheric blocking days, centered over the south-eastern part of Europe (Figure S12a). The spatial structure of
the Z500 anomalies, centered over the eastern part of Europe, leads to the advection of hot and dry air from the south or south-
eastern part of Europe. The large-scale atmospheric circulation associated with HWs over Romania, in July, is similar with
the spatial structure identified in June, both in the Z500 field (Figure S10b) as well as in the case of 2D atmospheric blocking
(Figure S12b). In August, the spatial structure of the Z500 field, associated with the occurrence of HWs over Romania, is
characterized by a wave-train like pattern of alternating Z500 anomalies, which extends from the eastern part of the U.S until
Eurasia (Figure S11c). Extreme HWs, in August, are associated with a low pressure system over the eastern part of the U.S.,
followed by positive Z500 anomalies over the western part of the central North Atlantic Ocean, negative Z500 anomalies
centered over the British Isles, and positive Z500 anomalies over the central and eastern parts of Europe. This wave-like
pattern suggests a stationary Rossby wave pattern, which is usually associated with heatwaves and droughts over the Eurasian
continent (Bakke et al., 2020; Barriopedro et al., 2011; Ionita et al., 2012; Schubert et al., 2014). As in the case of June and
July, HWs in August are also associated with an increased frequency of atmospheric blocking over the eastern part of Europe
(Figure S12c). The significant relationship between the HWDI and Z500 obtained via de composite map analysis is also



confirmed by the stability maps. June HWDI is stably and positively correlated with June Z500 over the eastern part of Europe,
centered over Romania (Figure 13a). The same pattern can be observed also when we compute the stability map between July
HWDI and July z500 (Figure 13c). In the case of August, the HWDI is stable and positively correlated with Z500 over a
region extended from the North Atlantic basin towards central and eastern part of Europe and negatively correlated with Z500
centered over the British isles and North Sea (Figure 13e). This dipole-structure is reminiscent of the East Atlantic
teleconnection pattern, which was found to have a significant influence on the variability of temperature and precipitation
over Europe, throughout the whole year (Gao et al., 2017). Based on the monthly stability maps identified in Figure 13, we
defined a Z500 index averaged over the stable regions (black squares in Figures 13a, 13c, and 13e) to analyze the interannual
variability of the Z500 over this regions in a long-term context. This analysis was motivated by the fact that it has been
suggested that the Z500 over central and western part of Europe has increased recently leading to an increase in the frequency
of HWs over these regions (Porebska and Zdunek, 2013; Tomczyk and Bednorz, 2016).  June Z500 index exhibits strong
interannual variability over the last 70 years, with the highest amplitudes since the beginning of 1990s (Figure 13b). Notably,
the highest value of this index was recorded in 2019, which is also the month with the longest June heatwave (Figure 2a).
Over the period 1990 – 2020 there is a significant increasing trend in the June Z500 averaged over the eastern part of Europe,
a trend which closely resembles the one observed for the June HWDI (Figure 2a). The results of the trend analysis for each
month and each analyzed period are given in Table S3. As in the case of June, July Z500 index exhibits also strong interannual
variability over the last 70 years and a significant increasing trend over since 1990's onward (Figure 13d). The highest values
of this index were recorded in 1954,1987, 1988, 2007, 2012 and 2015, respectively. July 2012 is also the month with the
longest July heatwave over the analyzed period (Figure 2c). The time series of August  Z500 index exhibits also strong
interannual variability over the last 70 years and a significant increasing trend over the period 1990 - 2020 (Figure 13f). The
highest value of this index was recorded in 1952, 1962, 1992, 2010, 2015, 2017 and 2019, respectively. August 2015 is also
the month with the longest July heatwave (Figure 2c). Overall, the time series of the monthly Z500 presents a strong
interannual variability and a significantly increasing trend starting with the beginning of the 1990's, which mirrors the trends
observed in the monthly HWDI (Figure 2). For July and August, the trend of the Z500 indices is significant for both analyzed
periods (i.e. 1950 – 200 and 1990 – 2020), while for June the trend is significant only when we consider the 1990 – 2020
period (Table S3).
**4 Conclusions**
One of the main conclusions of the recently published IPCC AR6 report (IPCC, 2021) was that "future heatwaves will last
longer and have higher temperatures". In this report (and the references therein) it has been shown that on a global scale there
is clear evidence of an increase in the number of warm nights and days and a decrease in the number of cold nights and days
(IPCC, 2021). Overall, the frequency of warm days (TX90p) has increased globally with small exceptions in the southern part
of South America (IPCC, 2021; Rusticucci et al., 2017). Over Europe an overall increase in the magnitude and frequency of
high maximum temperatures has been observed over central Europe (Lorenz et al., 2019; Tomczyk and Bednorz, 2016;
Twardosz and Kossowska-Cezak, 2013) and the southern-eastern part of Europe (Christidis et al., 2015; Croitoru et al., 2016a;
Croitoru and Piticar, 2013; Fioravanti et al., 2016; Malinovic-Milicevic et al., 2016). To extended the overview also for the
eastern part of Europe, in this study we provide an in-depth analysis of the trends and variability of hot and dry summers and





their combined effect (e.g. compound events), in the eastern part of Europe and their large-scale drivers, extending the analysis
period over more than 70 years (i.e. 1951 – 2020).
The main conclusions of this study can be summarized as follows: i) the length, spatial extent and frequency of HWs in
Romania has increased significantly over the last 70 years, for all summer months; ii) after the 1990's the rate of increase in
the frequency, length and spatial extent has significantly accelerated; iii) the longest and most extensive (in term of spatial
extent) HWs were observed in July 2012, August 2015 and June 2019; iv) no significant changes have been observed in the
drought conditions at country level; v) there is no significant increase in the compound events (e.g. hot and dry summers)
over the analyzed period and vi) the increased frequency of HWs especially after the 1990's could be partially explained by
an increase in the geopotential height over the eastern part of Europe.
A significant increase in the frequency of hot extremes has been found at country level, with the most affected regions being
in the north-western part and the Dobrogea region (Figure 4). Overall, an increase of the heat wave duration between 0.31
days/decade (in July) and 0.53 days/decade (in June) was observed (Table S1). The number of HWs started to increase in the
1990's reaching unprecedented length and spatial extent since 2000 until the end of the analyzed period (Figure 3). In terms
of drought variability, no significant changes have been found. The monthly SPI shows an inhomogeneous pattern of change,
with some regions experiencing drier condition (e.g. the north part of the country and Dobrogea region), while other regions
have become wetter over the last 70 years (e.g. eastern, central and western part of the country) (Figure 7). This
inhomogeneous pattern of change in the SPI is in agreement with previous studies which have shown that there is no spatial
consistency in the occurrence of droughts, based on the SPI, over Romania (Cheval et al., 2014; Ionita et al., 2016) and also
at European level (Ionita and Nagavciuc, 2021; Vicente-Serrano et al., 2021). The lack of homogeneity in the SPI variability,
has led to a strong variability when looking at the combined analysis of both hot and dry summers. The most active decades,
in terms of compound events, were 1951 -1960, 1991 – 2000 and 2011 – 2020. Throughout these periods, there were up to 4
combined hot and dry summers per decade, but their spatial distribution is different depending on the analyzed decade (Figure
9). Because the SPI has a very inhomogeneous spatial pattern, making it rather difficult to match the regions where also HWs
occur, the identified number of CHDs over the analyzed period was relatively small at country level, and no clear trend has
been observed in their frequency (not shown). Our results are not in agreement with other studies over different regions of
Europe (De Luca et al., 2020; Russo et al., 2019; Vogel et al., 2021), regarding the frequency of compound events (e.g. hot
and dry summers). All the aforementioned studies indicate that there is a significant increase in the frequency of CHD,
especially over the Mediterranean region. This discrepancy might be due to the fact the over the Mediterranean region there
is a homogenous trend regarding the drought conditions, thus hot and dry summers have a higher probability of occurring at
the same time.
The occurrence of HWs in the eastern part of Europe was related to anticyclonic conditions and a higher frequency of blocking
situations corroborated with daily maximum temperature anomalies up to 10°C (Figures S5 – S10). This is in agreement with
previous study for other regions (e.g. western part of Europe) which have shown that HWs tend to occur under the influence
of anticyclonic circulation, which is conductive to and intensification of the radiation flux and cloudless weather (Porebska
and Zdunek, 2013; Tomczyk et al., 2017; Tomczyk and Bednorz, 2016). The occurrence of HWs over the analyzed region is
stably correlated with the geopotential height centered over Romania and in the neighboring regions (Figure 13). The





geopotential height shows also an increase amplitude after the beginning of the 1990's, which follows the same temporal
variability as the HWDI index and the AREA index (Figure 2), thus supporting the finding that the increase in the number of
HWs over the last 2 decades could be explained, at least partially, by the increase in the regional geopotential height. Similar
results have been found also for the central and western part of Europe (Porebska and Zdunek, 2013; Tomczyk and Bednorz,
2016). In their study, Porebska and Zdunek (2013), have shown that heat waves over central part of Europe were often
associated with an increased frequency of blocking situations over the Atlantic Ocean and Eastern Europe. Similar results
have been found by Tomczyk and Bednorz (2016), which have shown that the occurrence of HWs in the central part of
Europe, was mainly driven with positive anomalies of the Z500 over the analyzed region. Thus, a possible explanation
regarding the increase in the frequency of HWs in Romania, over the past two decades, might be related to more frequent
blocking situations and an increase in the geopotential height over the analyzed region (Figure 13).
Our findings add more information to the recently published IPP report (IPCC, 2021), which states that there is an overall
global increase in the frequency of heatwaves and this pattern will continue in the future. This comprehensive analysis of the
variability and changes of heatwaves and droughts and their combined effect could be used to improve the adaptation strategies
to extreme events and to improve the resilience plans at country level.
















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

request.
***Financial support***. The article processing charges for this open access publication were covered by the Alfred Wegener
Institute, Helmholtz Centre for Polar and Marine Research (AWI).
***Competing interests***. The authors declare that they have no conflict of interest.























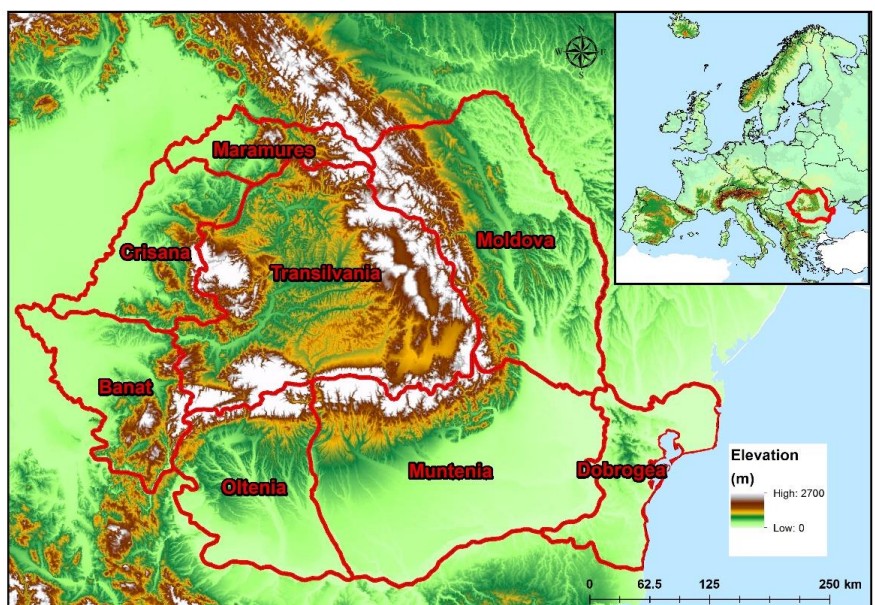

*Figure 1*. The topographic map of Romania and the location of the country at European level






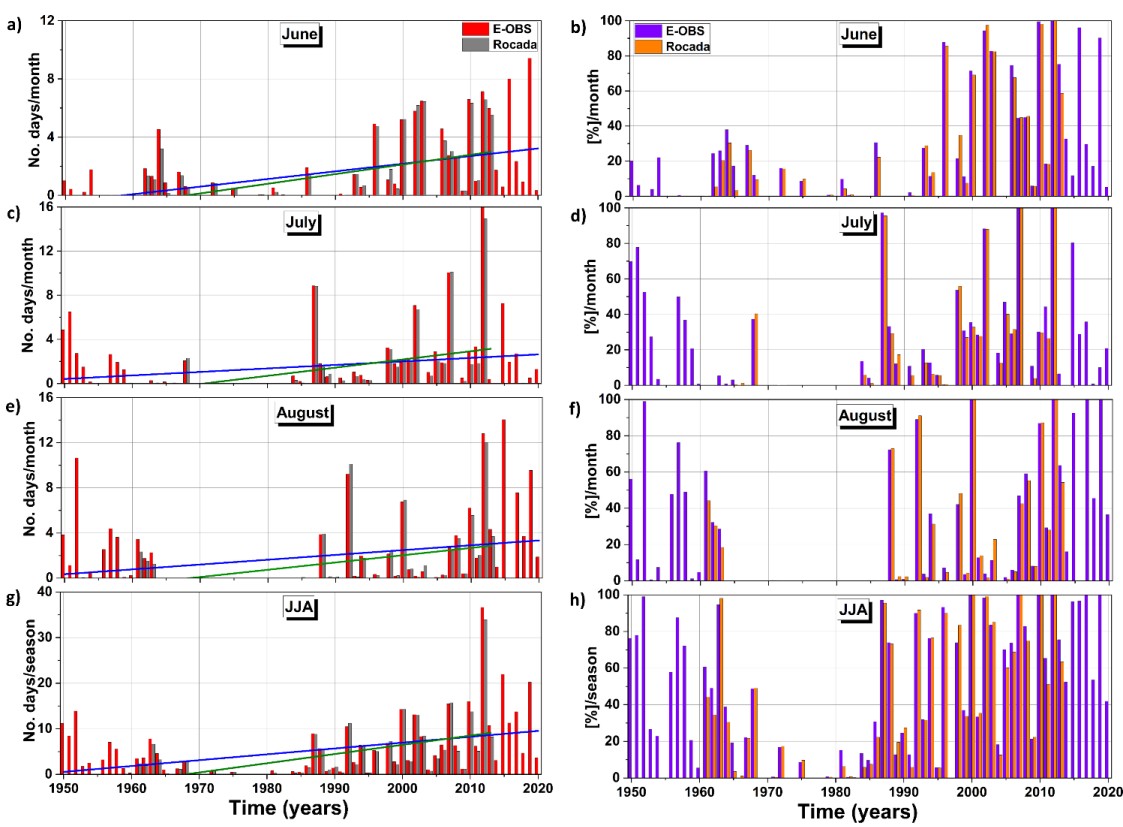

***Figure 2***. Monthly and seasonal temporal evolution of the summer heat waves duration (HWDI) averaged at country level (left column) and the temporal evolution of the percentage area (AREA) affected by heat waves (right column) over period 1950 – 2020: a) June HWDI; b) June AREA; c) July HWDI; d) July AREA; e) August HWDI; f) August AREA; g) Summer (JJA) HWDI and h) Summer (JJA) AREA. The orange lines indicate the time series obtained based on the E-OBA data set and the blue lines indicted the time series obtained based on the ROCADA dataset. The blue line indicates the linear trend line based on the E-OBS data and the green line represent the linear trend line based on the ROCDA data.






**Figure 3.** Decadal frequency of the number of summer heat waves (HWs) per decade over the last 70 years: a) 1951 – 1960;
b) 1961 – 1970; c)  1971 – 1980; d) 1981 – 1990; e) 1991 – 2000;
f) 2001 – 2010 and g) 2011 – 2020. Units: number of HWs/decade.


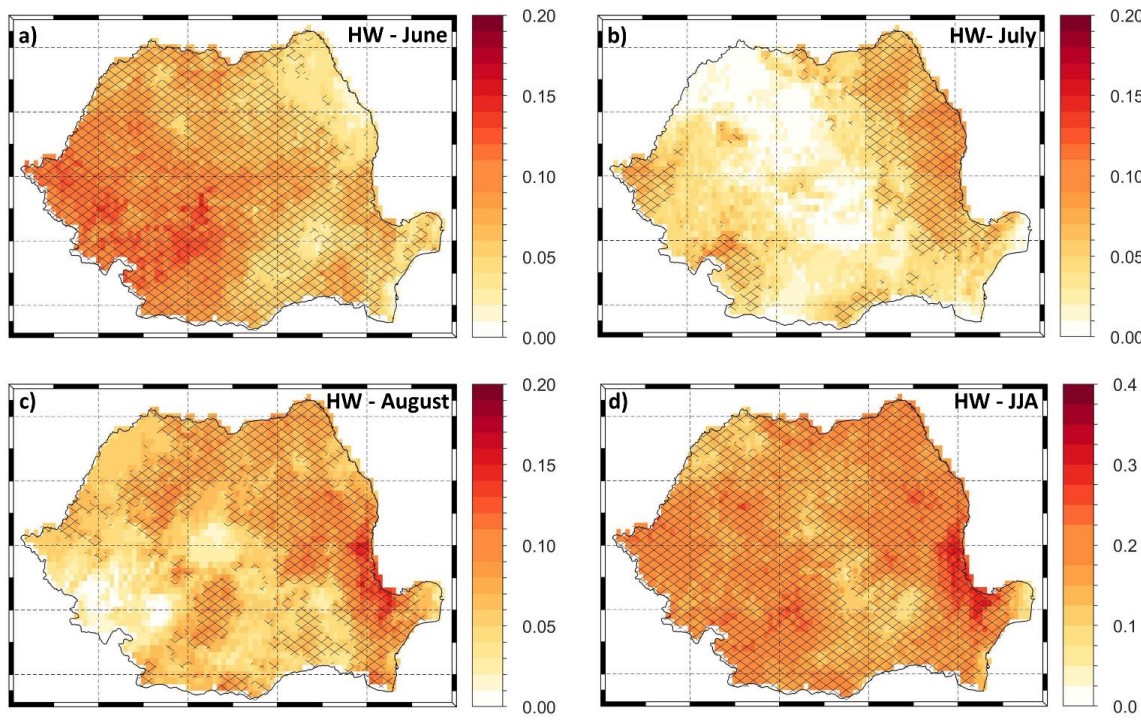

***Figure 4***. Linear trend of the numer heat waves for: a) june; b) July; c) August and d) JJA. Stipples indicate statistically significant trends. Units: number of HWs/decade. Analyzed period 1950 – 2020.











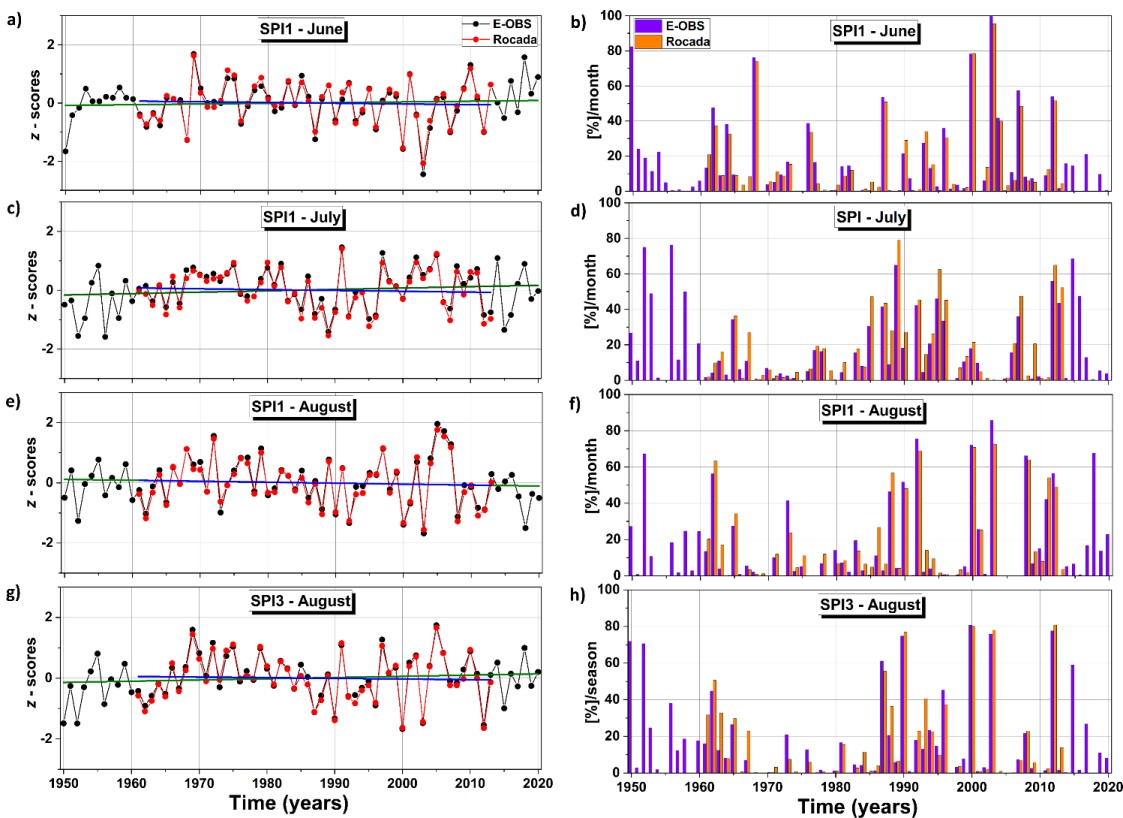

***Figure 5***. Monthly and seasonal temporal evolution of the SPI index averaged at country level (left column) and the temporal evolution of the percentage area (AREA) affected by drought conditions (SPI < -1) right column) over period 1950 – 2020: a) June SPI1; b) June drought AREA; c) July SPI 1 ; d) July drought AREA; e) August SPI1; f) August drought AREA; g) August SPI3 (indicator of dry/wet condition over the summer seasons) and h) August SPI3 drought AREA. The orange lines indicate the time series obtained based on the E-OBA data set and the blue lines indicted the time series obtained based on the ROCADA dataset. The blue line indicates the linear trend line based on the E-OBS data and the green line represent the linear trend line based on the ROCDA data.



![Figure 6 maps of decadal frequency of August SPI3 over Romania for seven decades]

**Figure 6**. Decadal frequency of August SPI3 over the last 70 years for the cases when
August SPI3 < -1: a) 1951 – 1960; b) 1961 – 1970; c)  1971 – 1980; d) 1981 – 1990;
e) 1991 – 2000; f) 2001 – 2010 and g) 2011 – 2020. Units: number of dry summers/decade.


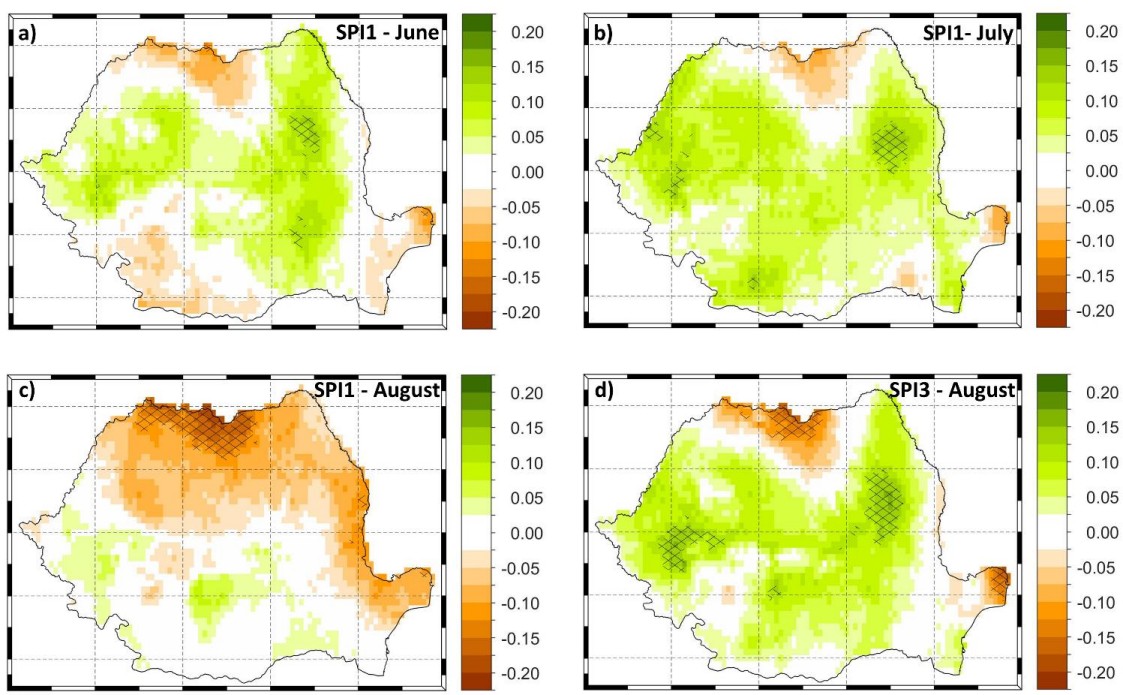

***Figure 7***. Linear trend of: a) June SPI1; b) July SPI1; c) August SPI1 and d) the Augsut SPI3. Stipples indicate
statistically significant trends. Units: number of z-scores/decade.
Analyzed period 1950 – 2020.














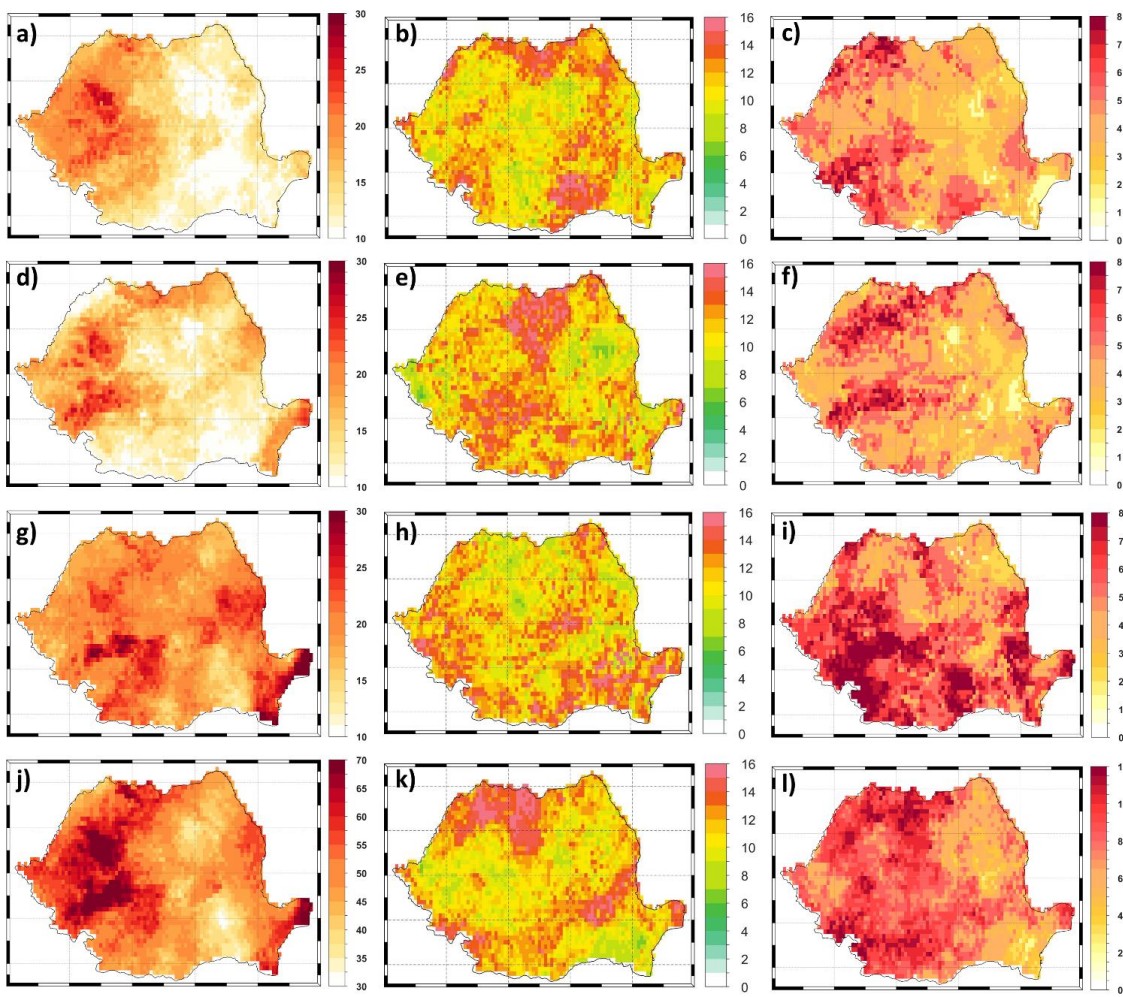

*Figure 8*. Frequency of monthly and seasonal HWs (first column), drought conditions (SPI<-1, second column) and compound hot and dry (CHD, third column over the whole analyzed period 1950 – 2020. a) June HWs; b) June SPI1; c) June CHD; d) July HWs; e) July SPI1; f) July CHD; g) August HWs; h) August SPI1; i) August CHD; j) Summer (JJA) HWs, k) August SPI3 and i) Summer (JJA) CHD. Units: HW (number of HWs/ 71 years), SPI (number of dry months/ 71 years) and CHD (number of CHDs/ 71 years).







*Figure 9.* Decadal frequency of the number of compound hot and dry summer (CHDs) per decade over the last 70 years:
a) 1951 – 1960; b) 1961 – 1970; c) 1971 – 1980; d) 1981 – 1990;
e) 1991 – 2000; f) 2001 – 2010 and g) 2011 – 2020. Units: number of CHDs/decade.

**Figure 10.** a) HWDI for July 2012; b) SPI1 for July 2012; c) Top-eight ranking of TX90p for July 2012 (1st means the, hottest (Tx90p) since 1950, 2nd signifies the second hottest, etc., and all ranks >8 are shown in white); d) CHD for June 2012 (the dark red color indicates the grid points affected by a CHD) and e) daily Z500 (contour lines) and TT850 anomalies (shaded colors) averaged over the period 25 - 30.07.2012.
Units: a) days/month; d) Z500 (m) and TT850 (°C). For d) the analyzed period is 1950–2020.




*Figure 11.* a) HWDI for August 2015; b) SPI1 for August 2015; c) Top-eight ranking of TX90p for August 2015 (1st means the, hottest (Tx90p) since 1950, 2nd signifies the second hottest, etc., and all ranks >8 are shown in white); d) CHD for August 2015 (the dark red color indicates the grid points affected by a CHD) and e) daily Z500 (contour lines) and TT850 anomalies (shaded colors) averaged over the period 28 - 31.08.2015.
Units: a) days/month; d) Z500 (m) and TT850 (°C). For d) the analyzed period is 1950–2020.



**Figure 12**. a) HWDI for June 2019; b) SPI1 for June 2019; c) Top-eight ranking of TX90p for June 2019 (1st means the, hottest (Tx90p) since 1950, 2nd signifies the second hottest, etc., and all ranks >8 are shown in white); d) CHD for June 2019 (the dark red color indicates the grid points affected by a CHD) and e) daily Z500 (contour lines) and TT850 anomalies (shaded colors) averaged over the period 10 - 14.06.2019.
Units: a) days/month; d) Z500 (m) and TT850 (°C). For d) the analyzed period is 1950–2020.






**Figure 13**. Stability maps of the correlation between monthly HWDI and monthly Z500 over the period 1950 – 2020 (left column) and the time series of monthly Z500 averaged over the black box in a), c) and e).

a) Stability map for June; b) The time series of June Z500 averaged overt the black box in a);

b) Stability map for July; d) the time series of July Z500 averaged overt the black box in c);

e) Stability map for August and f) the time series of August Z500 averaged overt the black box in e).

In a), c) and e) the regions where the correlation is positive for at least 80% of the 31-year windows are shaded with dark red (95 %), red (90 %), orange (85 %) and yellow (80 %). The corresponding regions where the correlation is significant, stable and negative, are shaded with dark blue (95 %), blue (90 %), green (85 %) and light green (80 %). The green (red) lines in b), d) and f) indicates the linear trend line of the monthly Z500 over the period 1950 – 2020 (1990 – 2020).
