# Peer review of "Hotspots for warm and dry summers in eastern Europe, with a focus on Romania"

_Natural Hazards and Earth System Sciences, 2021_

## Author Comment (AC1)

**GENERAL COMMENTS**

The paper presents an assessment of the spatio-temporal variability and trends of hot and dry summers, over the last fifty years, analyzing the physical mechanisms driving the occurrence of hot summers in Romania. For this, the heatwave duration index (HWDI), the Standardized and Precipitation index (SPI) and the compound hot and dry index (DHD) are computed for this region. I consider that this work is interesting, however, I also need to say, that I find the manuscript difficult to read, especially because the reader is constantly referred to supplementary material. Many of the figures in the supplementary material are necessary to follow the results. In this sense, I consider that a reorganization of the Methodology and Results sections is necessary. I think that an improvement of the paper is need previous to publication in order to reach the expected international standards requested by the journal.

R: Thank you for your constructive evaluation of our study. In the revised version of the manuscript we will consider all comments and suggestions and we will improved the manuscript accordingly (see detailed responses below).

**SPECIFIC COMMENTS**

1. Firstly, I think the authors are wrong in their attempt to extend their work on eastern Europe. All the calculations of the indices are made considering only data from Romania, and all the results shown in the manuscript are based on these indices. Although it is true that Romania is part of eastern Europe, the results obtained for just a country cannot be generalized to the complete eastern Europe. In my opinion this is an error, because from the title of the article the reader expects to find results referring to a much broader region. However, this fact does not detract from the value of the work, since Romania's geographical position, as well as its topographic characteristics, make it a very interesting region from a climatological point of view.

R: The title of the manuscript will be changed to reflect the analyzed region. More specifically the new title will be modified to: Hotspots for warm and dry summers in Romania

2. Other important point is about the use of the standardized precipitation index (SPI) to analyze drought events. I know that the SPI is a robust index widely used since it has a clear computation procedure and multi-scalar character. Nevertheless, the SPI only uses precipitation data to detect drought events. However, in the context of global warming is important to consider the effects of the temperature on drought. In this sense there is a new drought index, similar to SPI, that has the additional benefit of taking it into account. This is the Standardized Precipitation-Evapotranspiration Index (SPEI; Vicente Serrano et al., 2010), which combines the benefit of using the reference evapotranspiration with the simplicity, robustness, and the multi-scalar properties of the SPI. The increasing pattern of evaporation by global warming is not a negligible factor for drought assessment. So, SPEI is relatively better for drought monitoring compared with SPI. Taking this into account, I consider that the comparative study of regional applicability of these indices is highly required for suitable applications.

Vicente-Serrano, S. M., S. Beguería, and J. I. López-Moreno (2010), A multiscalar drought index sensitive to global warming: The Standardized Precipitation Evapotranspiration Index, J. Clim., 23, 1696–1718, doi:10.1175/2009JCLI29091.

R: The dispute SPI vs SPEI is not an easy one. When we drafted the study we wanted to use SPEI, but we decided for SPI for different reasons (see below). First of all, our aim was to compare a precipitation

based index with a temperature based index. Thus, we have chosen SPI on purpose, because if we would have use SPEI it would have meant comparing a temperature based index with another temperature related index and we wanted to avoid this comparison. SPEI is strongly affected by the global warming signal, thus we have tried avoiding using it in our analysis. Since SPEI is highly correlated with temperature it is also, to a certain degree, already an indicator for a compound event. Nevertheless, following the suggestions of all the reviewers involved in the review process of our manuscript in the revised version we are going to perform and show the same analysis by considering also SPEI.

3. About the use of ROCADA dataset, I don´t understand the advantage of using it because it has the same 0.1° x 0.1° spatial resolution than EOBs and shorter temporal cover.

R: We have added also the results of ROCADA dataset mainly because in previous studies we got complains that EOBS might not be suitable to make studies in Romania. But in the revised version of the manuscript we are going to remove the information and figures regarding the ROCADA dataset.

4. Page 1, lines 24-26: Authors literally conclude "that our study can help improve our understanding of the spatio-temporal variability of hot and dry summers, especially at the regional scale, as well as their driving mechanisms which might lead to a better predictability of these extreme events". I think that this cannot be a specific conclusion of this work. I suggest to change this with: "The results from this study can help improve our understanding of the spatio-temporal variability of hot and dry summer over Romania, as well as their driving mechanisms which might lead to a better predictability of these extreme events in the region."

R: The text will be modified as suggested by the reviewer.

5. Page 2, lines 82-88: A first summary about the main objective of the paper is made, and then this sound repeated in the description of the two main objectives. I suggest rewriting this by linking the two paragraphs.

R: The two paragraphs will be modified to make the text more clear and not repetitive.

6. Page 3, lines 97-98: Figure S1, which shows the temporal evolution of the heat wave duration index (HWDI) averaged for Romania for different durations, is introduced without explain the specific definition used for HWDI. Along with this, Figure S1 results are not relevant for the study, so I would suggest not showing this figure, especially considering the high number of figures in the manuscript (plus 12 figures in the supplementary material).

R: Figure S1 together with some other figures will be removed from the revised version of the manuscript.

7. Page 4, line 11: "(values <-1)" is referring to the values of SPI, which is cited later in the sentence. I suggest to eliminate this parentheses.

R: The text will be modified accordingly.

8. Page 4, line 121: the text in the parentheses is redundant. I suggest to change this with only (August SPI3 < -1).

R: The text will be modified as suggested.

9. Page 5, line 139: Figure 2 shows the HWDI averaged at the country level. ¿What is the meaning of that? ¿Is this the heatwave duration index averaged for Romania? If this is the case, the title of section 3.1 (summer heat waves in eastern Europe), must be changed by summer heat waves in Romania. I think that the complete analysis is centered in Romania, not using data from the rest of the countries of eastern Europe. So I think that even the title of the manuscript must be changed in order not to confuse to the reader.

R: Yes, in Figure 2 we have shown the heatwave duration index averaged for Romania. In the revised version of the manuscript we will changed the figure captions to make them more easy to follow and the title of each sub-section will also be modified to reflect the analyzed region, namely Romania.

10. Page 5, line 148: In table S1 results of the trend analysis for HWDI are shown. The trend analysis uses de Mann-Kendall test to detect the trend, but what method is used for trend estimation? All this information should be described in Methodology Section. A review of the methodology section is necessary.

R: The required information is going to be added in the Methodology Section.

11. Page 5, lines: 160-161: the average duration of HWs during the period 1950-1970 shows in Fig. 2g is lesser than 10 days.

R: The text will be changed.

12. Page 7, lines 220-223: This is repeated and was already explained in the methodology section.

R: The text will be removed.

13. Page 7, lines 228-235: I think that there are some errors in Figure 5. For example, in Figure 5a is stated June 2002 as one of the driest years. However I find from Fig. 5a that is 2003. ¿Is this correct? Similarly, from Figure 5g for SPI3, years 2002 and 2018 are stablished as driest summers. I find in this Figure that the years correspond to 2003 and 2012, respectively. Also, the quality of the Figures should be improved.

R: All the aforementioned Figures and years are going to be carefully checked in the revised version of the manuscript. Also we are going to improve the quality of the figures in the revised version of the manuscript.

14. Page 7, line 233: Again authors are referring to the eastern part of Europe. However, the analysis is just for Romania.

R: The text is going to be modified to reflect the studied region, namely Romania. These changes are going to be integrated throughout the whole manuscript.

15. Page 8, lines 264-268: I consider that this paragraph should be in Introduction section, and not in the results.

R: We agree with this suggestion. Thus, the aforementioned paragraph is going to be moved in the Introduction Section.

16. Page 9, line 307: The methodology used for ranking maps is explained in the supplementary material. I suggest to change it to the methodology section.

R: The methodology used for ranking maps is going to be moved in the main manuscript at the Methodology Section.

17. Page 9, line 318: In Figure S6 the location of the 2D atmospheric blocking is shown. The algorithm for the 2D atmospheric blocking index is also described in the supplementary material. I suggest to change it into the Methodology section.

R: The algorithm for the 2D atmospheric blocking index is going to be moved in the main manuscript at the Methodology Section.

18. Page 10, lines 316-324: In this paragraph is stablished that the pattern resulting from the atmospheric conditions is an increase in the number of hot days, especially in the southern and eastern regions of Romania. I cannot see this from figures 10e and S5. The evolution of the Tx anomaly (Figure S5) shows that this is maximum for the northern Romania.

R: In the revised version of the manuscript we will modify and improve the text following the reviewers suggestions and the text will be carefully checked to reflect the proper regions.

19. Page 11, line 374: I suggest to introduce a briefly description of the stability map methodology in the Methodology section. In summary, almost all the methodology used is explained in the supplementary material, which presents almost the same number of figures as the manuscript itself. Also, all the figures in the supplementary material are described in detail in the manuscript text, because they are supporting the results, so it is logical to think that they should be a specific part of the manuscript, and not supplementary material. In this sense, I consider that a restructuring of the manuscript is necessary.

R: We agree with the reviewer and the stability maps and the atmospheric blocking methodology is going to be moved in the main manuscript. Also some supplementary figures are going to be moved in the main manuscript in the revised version. Thus, the manuscript will be restructured as suggested by the reviewer.

20. Page 11, lines 363-368: If Figure S11 shows the composites maps of Z500 and wind for the years when the HWDI index (averaged for Romania) was > 5 days I consider not appropriate the figure caption for it, which establishes the occurrence of monthly heat waves in the central part of Europe.

R: The figure caption will be modified to reflect the study region.

21. Page 11, lines 380-382: The spatial structure of Z500 anomalies (Figure S11a) is indicating advection of air from the north-eastern part of Europe into Romania, not from the south-eastern.

R: The text will be modified as suggested.

22. Page 12, lines 420-427: Conclusion section begins with a paragraph with conclusions from other studies and for other regions in Europe. I think that this information could be appropriate to introduce the need of making this study in Romania, in the introduction section, but not here. Additionally, later in the conclusions the Figures showing the different results found are again mentioned. These figures have been previously described in detail in the Results section, so I consider that they must not be mentioned here again. Also, I suggest to change this section by Conclusion and Discussion section.

R: The manuscript will be substantially revised (e.g. by including SPEI in the analysis) and we are going to take into account the suggestions made by the reviewer regarding the Conclusion part and change the text accordingly.

**TECHNICAL CORRECTIONS**

Page 1, line 17: "2020and" should be "2020 and".

Page 1, line 18: "HWs" should be "Heat Waves (HWs)".

Page 2, line 40: "favors" should be "favour".

Page 2, lines 80-90: "The paper is structured as follow in Section 2 we give a detailed description of the data and methods used in this study. In Section 3 we…" should be "The paper is structured as follow: in Section 2 we give a detailed description of the data and methods used in this study; in Section 3 we…"

Page 5, line 158: "centuries" must be changed by "decades"

Page 7, line 216: To add (Figure 4c).

Page 7, line 217: to add (Figure 4d)

Page 7, lines 238-239: (Figure 6a) was already indicated at the beginning of the sentence.

Page 10, line 322: Figure S6d-S6g should be Figure S5d-S5g.

Page 10, line 335: In figure 11e contours indicating the countries are in white colour and this does not permit to visualize them correctly.

Page 11, line 370: "map s" should be "maps".

R: In the revised version of the manuscript all the technical corrections will be taken into account and the text and figures are going to be modified following the reviewer's suggestion.

Supplementary material:

In Figures from S6 to S12 the longitude and latitude labels must be indicated, at least in the maps of the lowest row.

In Figure S8, the contour lines indicating persistent atmospheric blocking system are very difficult to appreciate.

R: In the revised version of the manuscript we will try to improve all the figures tacking into account the aforementioned suggestions.

---

## Author Comment (AC2)

General Overview:

The authors analyze the hotspots for warm and dry summers in Romania using E-OBS and a regional dataset covering Romania. The authors intend to study the spatio-temporal variability and trends of hot and dry summers in the eastern part of Europe, focusing on Romania, between 1950 and 2020 and to study the relationship between the frequency of hot summers and the prevailing large-scale atmospheric circulation.

R: Thank you for your constructive evaluation of our study. In the revised version of the manuscript we will consider all comments and suggestions and we will improved the manuscript accordingly (see detailed responses below).

The manuscript fails in different aspects. Please find my major comments below:

- The authors should use in their analysis the SPEI over SPI. The SPEI is designed to consider both precipitation and potential evapotranspiration (PET) in determining drought. Thus, unlike the SPI, the SPEI captures the main impact of increased temperatures on water demand.

Vicente-Serrano et al. (2012)

R: The dispute SPI vs SPEI is not an easy one. When we drafted the study we wanted to use SPEI, but we decided for SPI for different reasons (see below). First of all, our aim was to compare a precipitation based index with a temperature based index. Thus, we have chosen SPI on purpose, because if we would have use SPEI it would have meant comparing a temperature based index with another temperature related index and we wanted to avoid this comparison. SPEI is strongly affected by the global warming signal, thus we have tried avoiding using it in our analysis. Since SPEI is highly correlated with temperature it is also, to a certain degree, already an indicator for a compound event. Nevertheless, following the suggestions of all the reviewers involved in the review process of our manuscript in the revised version we are going to perform and show the same analysis by considering also SPEI.

- I don´t understand the use of ROCADA database. One can argue that ROCADA use more weather station in the computation of the gridded dataset and therefore finer spatial scales will be resolved. However, throughout the manuscript it's not clear the different between EOBS and ROCADA neither the conclusion is different when using one or another. Therefore, I would go with the long -term dataset EOBS.

R: We have added also the results of ROCADA dataset mainly because in previous studies we got complains that EOBS might not be suitable to make studies in Romania. But in the revised version of the manuscript we are going to remove the information and figures regarding the ROCADA dataset.

- Section 3.3, this section intends to analyze the compound events in terms of hot and dry extremes. Are SPI < -1 really extreme? I don´t agree with the method used for defining compound event. They are only based on a month-to-month comparison and don´t go into further detail. What led to what? Pre-conditioning of soil moisture probably plays a role in the major Heat waves in the region. Have the authors though in using bi-variate methods to analyze the compound events? Or even to do a lag analysis between the dry and month summers?

R: Regarding the SPI threshold, this is a common threshold used in similar studies. We consider a threshold 0f -0.5 to be not a good indicator because it will include to many "dry" years in our analysis and a smaller than -1 would reduce dramatically the degrees of freedom for out analysis. We fully agree with the pre-conditioning. In this respect, in the revised version of the manuscript we are going to include lagged and in-phase correlation maps between the two indices (SPI and HWDI) to be able to better argue why the combination of different months in defining the compound events. Moreover in the revised version of the manuscript we are going to show also the probability of occurrence of risk, conditioned on SPI (SPEI) and HWDI.

Zscheischler and Fischer, 2020 : 10.1016/j.wace.2020.100270

Ribeiro at al., 2020 : 10.1016/j.wace.2020.100279

- Section 3.4., there is some lack of novelty in analyzing the synoptic meteorological patterns of the specific droughts years. No statistically significance is presented.

Sousa et al., 2021 : 10.1175/JCLI-D-20-0658.1

R: The anomaly maps for the case studies cannot have a significance filed because it is just a snapshot for one event in time, thus we cannot perform any statistical significance. For the composite maps the significance of the anomalies is actually plotted in the figures (see figures S11 and S12 for example). It was a misfortune from our side that we did not write that clearly in the figure caption.

Regarding the comment "there is some lack of novelty in analyzing the synoptic meteorological patterns of the specific droughts years", we do not agree with it. Each region has it's own particularities, thus the large-scale drivers have different spatial structures. Yes, of course a heat wave will most probably be driven by a blocking system, but this doesn't mean that we have fixed and fully closed the issue of analyzing drivers of extreme events. Moreover, we have added also the stability maps in our manuscript (Figure 13) to further add some new info regarding the relationship between heatwaves and their drivers and we consider this is a new way to study this kind of relationship.

Therefore, all the changes need to be made, in order to the paper goes for a second round of revision.

R: In the revised version of the manuscript we are going to try to take into account all the aforementioned suggestions.

---

## Author Comment (AC3)

Reviewer 3

The manuscript does not have enough scientific merit to be published in the journal. It does not provide significantly new information which go beyond the current state of the art. It is descriptive and does not add new elements in current understanding of compound extremes in the area. Further a few parts are inconsistent and also not scientifically sound. Thus, I suggest rejection. A detailed review is provided below.

R: We do not agree with these comments. For sure there is room for improvement of the manuscript and we thank the reviewers for helping us in this respect, but we do not agree that our manuscript does not provide significantly new information which go beyond the current state of the art. We are going to support our argument by answering point by point the reviewer's comment below.

The title is misleading as it only deals with Romania, not Eastern Europe.

R: In the title of the manuscript it's very clearly stated that the manuscript is focused on Romania. Nevertheless, the title will be change to reflect the analyzed region.

The abstract mentions that compound extremes are considered. However, it only reports on changes in extreme temperature and precipitation/drought separately. There is a lot of methodological papers out in the literature that deal with compound extremes, how they are modelled using sophisticated methods.

R: We disagree with this comment. Both changes in temperature and precipitation are analyzed individually and also combined (see Section 3.3). Our aim was to analyze if there are any changes in the joint frequency of warm and dry spells, and this has been analyzed in detail in Section 3.3.

Nevertheless, in the revised version of the manuscript we are going to improve the methodology for computing the compound events. In this respect, in the revised version of the manuscript we are going to include lagged and in-phase correlation maps between the two indices (SPI and HWDI) to be able to better argue why the combination of different months in defining the compound events. Moreover in the revised version of the manuscript we are going to show also the probability of occurrence of risk, conditioned on SPI (SPEI) and HWDI.

Introduction: There is a lot of literature that deals with extremes in southeastern Europe, including Romania, for instance also in the form of reviews:

Kuglitsch, F. et al. 2010: Heat Wave Changes in the Eastern Mediterranean since 1960, Geophys. Res. Lett., 37, L04802.

Ulbrich, U. et al. 2013: Climate of the Mediterranean: synoptic patterns, temperature, precipitation, winds and their extremes. Future Climate Projections. In: Regional Assessment of Climate Change in the Mediterranean: A. Navarra, L. Tubiana (eds.), Springer

R: We do not really think that the suggested papers are state of the art papers regarding the occurrence of extreme events over Romania. The papers indicated by the reviewer are rather old and do just a superficial analysis of the extreme events in Romania. For example the paper of Kuglitch et al., 2010 focuses on a period between 1960 - 2006. From a temporal point of view this is rather old, and new analysis is always indicated, especially due to the fact that most of the extreme years in term of extreme temperature and precipitation have occurred over the last 20 years. Moreover, in the aforementioned

study a limited number of station covering Romania are used, which do not really give a proper overview of the complex climatology of the country. Moreover, the aforementioned comment of the reviewer implies more or less that scientist should stop doing regional studies, just because there are continental and/or global studies.

The introduction does not provide a clear justification why this work is needed, does not show gaps in current understanding and does not formulate a clear hypothesis.

R: In the revised version of the manuscript we are going to improve the introduction part in order to make the justification of the paper more clear.

The choice of more than 5 days defining a heatwave is not objectively based (lines 109/110). My suggestion would be to consult the latest literature that deal with more objective measures how heatwaves are defined.

R: Our choice of 5 days was based on the recommended thresholds for the regions surrounding Romania. Moreover, we also followed the recommendations of the Expert Team on Climate Change Detection and Indices (ETCCDI).

Heatwave results reported (lines 149-150, 4 to 5 days) are in disagreement with the definition provided in lines 109/110 that state more than 5 days.

R: In the respective line we speak about an average at country level, meaning that the number is an average over a certain number of grid points. Some grid points might fulfil the heatwave criteria, some not, thus when you average them you will get a number averaged over a large region.

Lines 160/161, there is an overlap of having the year 1970 in both periods. In addition, the period 1970-1985 has a different length compared to others that makes the comparison difficult.

R: The text will be modified.

SPI is not the most appropriate measure for drought. For the area, SPEI is a better index that combines temperature and precipitation.

R: We do not really agree with this comment. In this study we compare a precipitation based index with a temperature based index. We have chosen SPI on purpose, because if we would have use SPEI it would have meant comparing a temperature based index with another temperature related index and we wanted to avoid this comparison. SPEI is strongly affect by the global warming signal, thus we have tried avoiding using it in our analysis. Since SPEI is highly correlated with temperature is to a certain degree also already an indicator for a compound event. Nevertheless, in the revised version of the manuscript we are going to perform the same analysis by considering also SPEI.

For a review please consult Raible et al. (2017): Drought indices revisited – improving and testing of drought indices in a simulation of the last two millennia for Europe, Tellus A: Dynamic Meteorology and Oceanography, 69, 1287492.

R: We do not think that this paper is actually relevant for giving a clear suggestion which index is optimal, mainly because it's a modeling study and most of the models have issues in properly represent the potential evapotranspiration which is an essential component in computing SPEI. We think the choice of the drought index should reflect what the authors want to analyze. Out aim was to analyze changes in extreme temperature and extreme precipitation and we have tried to identify the proper indices to do so. A more indicate paper in this respect would be Stagge et al., 2017. Based on the analysis of Stagge et al.(2017) there are no significant difference between SPEI and SPI over our analyzed region (Figure 2c in their paper). Nevertheless, following the suggestions of all the reviewers involved in the review process of our manuscript in the revised version we are going to perform and show the same analysis by considering also SPEI.

The comparison between EOBs and ROCADA does not provide new evidence, it could be skipped and the analysis could be concentrated on EOBs.

R: We have added also the results of ROCADA dataset mainly because in previous studies we got complain that EOBS might not be suitable to make studies in Romania. But in the revised version of the manuscript we are going to remove the information and figures regarding the ROCADA dataset.

Sentence on lines 122-123 is not clear.

R: The text will be modified and improved.

The manuscript states at various places "statistical significant" changes. However, no information on the underlying statistics to test significance is provided. A few maps show significant areas related to trends, however it is missing how those regions are calculated.

R: This was a misfortune form our part. In the revised version of the manuscript we are going to add in the Methodology text all the statistical test and the associated references used throughout the manuscript.

Further, the synoptical maps do not have a field significance information and thus they are difficult to interpret.

R: The anomaly maps for the case studies cannot have a significance field because it is just a snapshot for one event in time, thus we cannot perform any statistical significance.

For the composite maps the significance of the anomalies is actually plotted in the figures (see figures S11 and S12 for example). Again it was a misfortune from our side that we did not write that clearly in the figure caption.

Further, the maps are not unexpected and the processes that lead to drought or heat extremes are well documented in the literature elsewhere.

R: We really disagree with this comment. The fact that processes that lead to drought/heatwaves are well documented in literature, does not mean that scientist should stop doing this kind of research. Each region has it's own particularities. Yes, of course a heat wave will most probably be driven by a blocking system, but this doesn't mean that we have fixed and fully closed the issue of analyzing drivers of extreme events. Again, it seems to be a very accepted approach for most of the already published studies, but for the current study doesn't seem to be accepted. Moreover, we have added also the stability maps in our manuscript (Figure 13) to further add some new info regarding the relationship between heatwaves and

their drivers and we consider this is a new way to study this kind of relationship. Thus, we think this comment does not really reflect the findings and analysis from our manuscript.

The conclusions include information from the introduction and duplicate the results. As such, a lot of information is irrelevant and the last paragraph is not a conclusion from the analysis shown.

R: The text will be modified, improved and adjusted to the new figures which are going to be produced throughout the review process.

---

## Author Response (AR1)

**Reviewer 1**

**GENERAL COMMENTS**

The paper presents an assessment of the spatio-temporal variability and trends of hot and dry summers, over the last fifty years, analyzing the physical mechanisms driving the occurrence of hot summers in Romania. For this, the heatwave duration index (HWDI), the Standardized and Precipitation index (SPI) and the compound hot and dry index (DHD) are computed for this region. I consider that this work is interesting, however, I also need to say, that I find the manuscript difficult to read, especially because the reader is constantly referred to supplementary material. Many of the figures in the supplementary material are necessary to follow the results. In this sense, I consider that a reorganization of the Methodology and Results sections is necessary. I think that an improvement of the paper is need previous to publication in order to reach the expected international standards requested by the journal.

R: Thank you for your constructive evaluation of our study. In the revised version of the manuscript we have consider all comments and suggestions and we have improved the manuscript accordingly (see detailed responses below).

**SPECIFIC COMMENTS**

1. Firstly, I think the authors are wrong in their attempt to extend their work on eastern Europe. All the calculations of the indices are made considering only data from Romania, and all the results shown in the manuscript are based on these indices. Although it is true that Romania is part of eastern Europe, the results obtained for just a country cannot be generalized to the complete eastern Europe. In my opinion this is an error, because from the title of the article the reader expects to find results referring to a much broader region. However, this fact does not detract from the value of the work, since Romania's geographical position, as well as its topographic characteristics, make it a very interesting region from a climatological point of view.

R: The title of the manuscript will has been changed to reflect the analyzed region. More specifically the new title is: Hotspots for warm and dry summers in Romania

2. Other important point is about the use of the standardized precipitation index (SPI) to analyze drought events. I know that the SPI is a robust index widely used since it has a clear computation procedure and multi-scalar character. Nevertheless, the SPI only uses precipitation data to detect drought events. However, in the context of global warming is important to consider the effects of the temperature on drought. In this sense there is a new drought index, similar to SPI, that has the additional benefit of taking it into account. This is the Standardized Precipitation-Evapotranspiration Index (SPEI; Vicente Serrano et al., 2010), which combines the benefit of using the reference evapotranspiration with the simplicity, robustness, and the multi-scalar properties of the SPI. The increasing pattern of evaporation by global warming is not a negligible factor for drought assessment. So, SPEI is relatively better for drought monitoring compared with SPI. Taking this into account, I consider that the comparative study of regional applicability of these indices is highly required for suitable applications.

Vicente-Serrano, S. M., S. Beguería, and J. I. López-Moreno (2010), A multiscalar drought index sensitive to global warming: The Standardized Precipitation Evapotranspiration Index, J. Clim., 23, 1696–1718, doi:10.1175/2009JCLI29091.

R: In the revised version of the manuscript we have replaced the SPI with SPEI and the text and the results have been described accordingly.

3. About the use of ROCADA dataset, I don´t understand the advantage of using it because it has the same 0.1° x 0.1° spatial resolution than EOBs and shorter temporal cover.

R: We have added also the results of ROCADA dataset mainly because in previous studies we got complains that EOBS might not be suitable to make studies in Romania. But in the revised version of the manuscript have to removed the information and figures regarding the ROCADA dataset.

4. Page 1, lines 24-26: Authors literally conclude "that our study can help improve our understanding of the spatio-temporal variability of hot and dry summers, especially at the regional scale, as well as their driving mechanisms which might lead to a better predictability of these extreme events". I think that this cannot be a specific conclusion of this work. I suggest to change this with: "The results from this study can help improve our understanding of the spatio-temporal variability of hot and dry summer over Romania, as well as their driving mechanisms which might lead to a better predictability of these extreme events in the region."

R: The text has been modified as suggested by the reviewer.

5. Page 2, lines 82-88: A first summary about the main objective of the paper is made, and then this sound repeated in the description of the two main objectives. I suggest rewriting this by linking the two paragraphs.

R: The two paragraphs have been modified to make the text more clear and not repetitive.

6. Page 3, lines 97-98: Figure S1, which shows the temporal evolution of the heat wave duration index (HWDI) averaged for Romania for different durations, is introduced without explain the specific definition used for HWDI. Along with this, Figure S1 results are not relevant for the study, so I would suggest not showing this figure, especially considering the high number of figures in the manuscript (plus 12 figures in the supplementary material).

R: Figure S1 together with some other figures have been removed from the revised version of the manuscript.

7. Page 4, line 11: "(values <-1)" is referring to the values of SPI, which is cited later in the sentence. I suggest to eliminate this parentheses.

R: The text has been modified accordingly.

8. Page 4, line 121: the text in the parentheses is redundant. I suggest to change this with only (August SPI3 < -1).

R: The text has been modified as suggested.

9. Page 5, line 139: Figure 2 shows the HWDI averaged at the country level. ¿What is the meaning of that? ¿Is this the heatwave duration index averaged for Romania? If this is the case, the title of section 3.1 (summer heat waves in eastern Europe), must be changed by summer heat waves in Romania. I think that the complete analysis is centered in Romania, not using data from the rest of the countries of eastern Europe. So I think that even the title of the manuscript must be changed in order not to confuse to the reader.

R: Yes, in Figure 2 we have shown the heatwave duration index averaged for Romania. In the revised version of the manuscript we have changed the figure captions to make them more easy to follow and the title of each sub-section will also be modified to reflect the analyzed region, namely Romania.

10. Page 5, line 148: In table S1 results of the trend analysis for HWDI are shown. The trend analysis uses de Mann-Kendall test to detect the trend, but what method is used for trend estimation? All this information should be described in Methodology Section. A review of the methodology section is necessary.

R: The required information has been added in the Methodology Section.

11. Page 5, lines: 160-161: the average duration of HWs during the period 1950-1970 shows in Fig. 2g is lesser than 10 days.

R: The text has been changed.

12. Page 7, lines 220-223: This is repeated and was already explained in the methodology section.

R: The corresponding paragraph has been removed from the revised version of the manuscript.

13. Page 7, lines 228-235: I think that there are some errors in Figure 5. For example, in Figure 5a is stated June 2002 as one of the driest years. However I find from Fig. 5a that is 2003. ¿Is this correct? Similarly, from Figure 5g for SPI3, years 2002 and 2018 are stablished as driest summers. I find in this Figure that the years correspond to 2003 and 2012, respectively. Also, the quality of the Figures should be improved.

R: All the aforementioned Figures and years have been carefully checked in the revised version of the manuscript. Since we have used SPEI instead of SPI the text has been substantially changed for this whole section. Also have tried to improve the quality of the figures in the revised version of the manuscript.

14. Page 7, line 233: Again authors are referring to the eastern part of Europe. However, the analysis is just for Romania.

R: The text has been modified to reflect the studied region, namely Romania. These changes have been integrated throughout the whole manuscript.

15. Page 8, lines 264-268: I consider that this paragraph should be in Introduction section, and not in the results.

R: We agree with this suggestion. Thus, the aforementioned paragraph has been moved in the Introduction Section.

16. Page 9, line 307: The methodology used for ranking maps is explained in the supplementary material. I suggest to change it to the methodology section.

R: The methodology used for ranking maps is going has been moved in the main manuscript in the Methodology Section.

17. Page 9, line 318: In Figure S6 the location of the 2D atmospheric blocking is shown. The algorithm for the 2D atmospheric blocking index is also described in the supplementary material. I suggest to change it into the Methodology section.

R: The algorithm for the 2D atmospheric blocking index has been moved in the main manuscript in the Methodology Section.

18. Page 10, lines 316-324: In this paragraph is stablished that the pattern resulting from the atmospheric conditions is an increase in the number of hot days, especially in the southern and eastern regions of Romania. I cannot see this from figures 10e and S5. The evolution of the Tx anomaly (Figure S5) shows that this is maximum for the northern Romania.

R: In the revised version of the manuscript we will modify and improve the text following the reviewers suggestions and the text will be carefully checked to reflect the proper regions.

19. Page 11, line 374: I suggest to introduce a briefly description of the stability map methodology in the Methodology section. In summary, almost all the methodology used is explained in the supplementary material, which presents almost the same number of figures as the manuscript itself. Also, all the figures in the supplementary material are described in detail in the manuscript text, because they are supporting the results, so it is logical to think that they should be a specific part of the manuscript, and not supplementary material. In this sense, I consider that a restructuring of the manuscript is necessary.

R: We agree with the reviewer and the stability maps and the atmospheric blocking methodology has been moved in the main manuscript. Also some supplementary figures have been moved in the main manuscript in the revised version. Thus, the manuscript has been restructured as suggested by the reviewer.

20. Page 11, lines 363-368: If Figure S11 shows the composites maps of Z500 and wind for the years when the HWDI index (averaged for Romania) was > 5 days I consider not appropriate the figure caption for it, which establishes the occurrence of monthly heat waves in the central part of Europe.

R: The figure caption has been modified to reflect the study region.

21. Page 11, lines 380-382: The spatial structure of Z500 anomalies (Figure S11a) is indicating advection of air from the north-eastern part of Europe into Romania, not from the south-eastern.

R: The text has been modified as suggested.

22. Page 12, lines 420-427: Conclusion section begins with a paragraph with conclusions from other studies and for other regions in Europe. I think that this information could be appropriate to introduce the need of making this study in Romania, in the introduction section, but not here. Additionally, later in the conclusions the Figures showing the different results found are again mentioned. These figures have been previously described in detail in the Results section, so I consider that they must not be mentioned here again. Also, I suggest to change this section by Conclusion and Discussion section.

R: The manuscript has been substantially revised (e.g. by including SPEI in the analysis) and have took into account the suggestions made by the reviewer regarding the Conclusion part and changed the text accordingly.

**TECHNICAL CORRECTIONS**

Page 1, line 17: "2020and" should be "2020 and".

Page 1, line 18: "HWs" should be "Heat Waves (HWs)".

Page 2, line 40: "favors" should be "favour".

Page 2, lines 80-90: "The paper is structured as follow in Section 2 we give a detailed description of the data and methods used in this study. In Section 3 we…" should be "The paper is structured as follow: in Section 2 we give a detailed description of the data and methods used in this study; in Section 3 we…"

Page 5, line 158: "centuries" must be changed by "decades"

Page 7, line 216: To add (Figure 4c).

Page 7, line 217: to add (Figure 4d)

Page 7, lines 238-239: (Figure 6a) was already indicated at the beginning of the sentence.

Page 10, line 322: Figure S6d-S6g should be Figure S5d-S5g.

Page 10, line 335: In figure 11e contours indicating the countries are in white colour and this does not permit to visualize them correctly.

Page 11, line 370: "map s" should be "maps".

R: In the revised version of the manuscript all the technical corrections have been taken into account and the text and figures are going to be modified following the reviewer's suggestion.

Supplementary material:

In Figures from S6 to S12 the longitude and latitude labels must be indicated, at least in the maps of the lowest row.

In Figure S8, the contour lines indicating persistent atmospheric blocking system are very difficult to appreciate.

R: In the revised version of the manuscript we have tried to improve all the figures by tacking into account the aforementioned suggestions.

**Reviewer 2**

General Overview:

The authors analyze the hotspots for warm and dry summers in Romania using E-OBS and a regional dataset covering Romania. The authors intend to study the spatio-temporal variability and trends of hot and dry summers in the eastern part of Europe, focusing on Romania, between 1950 and 2020 and to study the relationship between the frequency of hot summers and the prevailing large-scale atmospheric circulation.

R: Thank you for your constructive evaluation of our study. In the revised version of the manuscript we have consider all comments and suggestions and we have improved the manuscript accordingly (see detailed responses below).

The manuscript fails in different aspects. Please find my major comments below:

- The authors should use in their analysis the SPEI over SPI. The SPEI is designed to consider both precipitation and potential evapotranspiration (PET) in determining drought. Thus, unlike the SPI, the SPEI captures the main impact of increased temperatures on water demand.

Vicente-Serrano et al. (2012)

R: In the revised version of the manuscript we have replaced the SPI with SPEI and the text and the results have been described accordingly.

- I don´t understand the use of ROCADA database. One can argue that ROCADA use more weather station in the computation of the gridded dataset and therefore finer spatial scales will be resolved. However, throughout the manuscript it's not clear the different between EOBS and ROCADA neither the conclusion is different when using one or another. Therefore, I would go with the long -term dataset EOBS.

R: We have added also the results of ROCADA dataset mainly because in previous studies we got complains that EOBS might not be suitable to make studies in Romania. But in the revised version of the manuscript have to removed the information and figures regarding the ROCADA dataset.

- Section 3.3, this section intends to analyze the compound events in terms of hot and dry extremes. Are SPI < -1 really extreme? I don´t agree with the method used for defining compound event. They are only based on a month-to-month comparison and don´t go into further detail. What led to what? Pre-conditioning of soil moisture probably plays a role in the major Heat waves in the region. Have the authors though in using bi-variate methods to analyze the compound events? Or even to do a lag analysis between the dry and month summers?

R: Regarding the SPI threshold, this is a common threshold used in similar studies. We consider a threshold of -0.5 to be not a good indicator because it will include to many "dry" years in our analysis and a smaller than -1 would reduce dramatically the degrees of freedom for out analysis. We fully agree with the pre-conditioning. In this respect, in the revised version of the manuscript we have included lagged and in-phase correlation maps between the two indices (SPEI and HWDI) to be able to better argue why the combination of different months in defining the compound events.

Zscheischler and Fischer, 2020 : 10.1016/j.wace.2020.100270

Ribeiro at al., 2020 : 10.1016/j.wace.2020.100279

- Section 3.4., there is some lack of novelty in analyzing the synoptic meteorological patterns of the specific droughts years. No statistically significance is presented.

Sousa et al., 2021 : 10.1175/JCLI-D-20-0658.1

R: The anomaly maps for the case studies cannot have a significance filed because it is just a snapshot for one event in time, thus we cannot perform any statistical significance. For the composite maps the significance of the anomalies is actually plotted in the figures (see Figures 13 and 14). It was a misfortune from our side that we did not write that clearly in the figure caption.

Regarding the comment "there is some lack of novelty in analyzing the synoptic meteorological patterns of the specific droughts years", we do not agree with it. Each region has it's own particularities, thus the large-scale drivers have different spatial structures. Yes, of course a heat wave will most probably be driven by a blocking system, but this doesn't mean that we have fixed and fully closed the issue of analyzing drivers of extreme events. Moreover, we have added also the stability maps in our manuscript (Figure 13) to further add some new info regarding the relationship between heatwaves and their drivers and we consider this is a new way to study this kind of relationship.

Therefore, all the changes need to be made, in order to the paper goes for a second round of revision.

R: In the revised version of the manuscript we have tried to take into account all the aforementioned suggestions and modified the text and figure following the reviewer's suggestions/comments.

**Reviewer 3**

The manuscript does not have enough scientific merit to be published in the journal. It does not provide significantly new information which go beyond the current state of the art. It is descriptive and does not add new elements in current understanding of compound extremes in the area. Further a few parts are inconsistent and also not scientifically sound. Thus, I suggest rejection. A detailed review is provided below.

R: We do not agree with these comments. For sure there is room for improvement of the manuscript and we thank the reviewers for helping us in this respect, but we do not agree that our manuscript does not provide significantly new information which go beyond the current state of the art. We are going to support our argument by answering point by point the reviewer's comment below.

The title is misleading as it only deals with Romania, not Eastern Europe.

R: The title has been changed to reflect the analyzed region. More specifically the new title is: Hotspots for warm and dry summers in Romania.

The abstract mentions that compound extremes are considered. However, it only reports on changes in extreme temperature and precipitation/drought separately. There is a lot of methodological papers out in the literature that deal with compound extremes, how they are modelled using sophisticated methods.

R: We disagree with this comment. Both changes in temperature and precipitation are analyzed individually and also combined (see Section 3.3). Our aim was to analyze if there are any changes in the joint frequency of warm and dry spells, and this has been analyzed in detail in Section 3.3.

Nevertheless, in the revised version of the manuscript we have improved the methodology for computing the compound events. In this respect, in the revised version of the manuscript we have included lagged and in-phase correlation maps between the two indices (SPEI and HWDI) to be able to better argue why the combination of different months in defining the compound events (see Figure 8 in the revised version of the manuscript). Moreover in the revised version of the manuscript we have included also conditional probability maps for the two analyzed indices (see Figure 9 in the revised version of the manuscript).

Introduction: There is a lot of literature that deals with extremes in southeastern Europe, including Romania, for instance also in the form of reviews:

Kuglitsch, F. et al. 2010: Heat Wave Changes in the Eastern Mediterranean since 1960, Geophys. Res. Lett., 37, L04802.

Ulbrich, U. et al. 2013: Climate of the Mediterranean: synoptic patterns, temperature, precipitation, winds and their extremes. Future Climate Projections. In: Regional Assessment of Climate Change in the Mediterranean: A. Navarra, L. Tubiana (eds.), Springer

R: We do not really think that the suggested papers are state of the art papers regarding the occurrence of extreme events over Romania. The papers indicated by the reviewer are rather old and do just a superficial analysis of the extreme events in Romania. For example the paper of Kuglitch et al., 2010 focuses on a period between 1960 - 2006. From a temporal point of view this is rather old, and new analysis is always indicated, especially due to the fact that most of the extreme years in term of extreme temperature and precipitation have occurred over the last 20 years. Moreover, in the aforementioned study a limited number of station covering Romania are used, which do not really give a proper overview of the complex climatology of the country. Moreover, the aforementioned comment of the reviewer implies more or less that scientist should stop doing regional studies, just because there are continental and/or global studies. Below please find just some recent papers which are dealing with regional and event based studies (this are the most recent ones just to give some examples):

Bakke, S. J., Ionita, M. and Tallaksen, L. M.: The 2018 northern European hydrological drought and its drivers in a historical perspective, Hydrol. Earth Syst. Sci., 24(11), 5621–5653, doi:10.5194/hess-24-5621-2020, 2020.

Duchez, A., Frajka-Williams, E., Josey, S. A., Evans, D. G., Grist, J. P., Marsh, R., McCarthy, G. D., Sinha, B., Berry, D. I. and Hirschi, J. J. M.: Drivers of exceptionally cold North Atlantic Ocean temperatures and their link to the 2015 European heat wave, Environ. Res. Lett., 11(7), doi:10.1088/1748-9326/11/7/074004, 2016.

Marengo, J. A., Ambrizzi, T., Barreto, N., Cunha, A. P., Ramos, A. M., Skansi, M., Molina Carpio, J. and Salinas, R.: The heat wave of October 2020 in central South America, Int. J. Climatol., n/a(n/a), doi:https://doi.org/10.1002/joc.7365, n.d.

McCarthy, M., Christidis, N., Dunstone, N., Fereday, D., Kay, G., Klein-Tank, A., Lowe, J., Petch, J., Scaife, A. and Stott, P.: Drivers of the UK summer heatwave of 2018, Weather, 74(11), 390–396, doi:https://doi.org/10.1002/wea.3628, 2019.

Overland, J. E. and Wang, M.: The 2020 Siberian heat wave, Int. J. Climatol., 41(S1), E2341–E2346, doi:https://doi.org/10.1002/joc.6850, 2021.

Sinclair, V. A., Mikkola, J., Rantanen, M. and Räisänen, J.: The summer 2018 heatwave in Finland, Weather, 74(11), 403–409, doi:https://doi.org/10.1002/wea.3525, 2019.

Vautard, R., Boucher, O., Geert, P. ), Van Oldenborgh, J., Otto, F., Haustein, K., Vogel, M. M., Seneviratne, S. I., Soubeyroux, J.-M., Schneider, M. and Drouin, A.: Human contribution to the record-breaking July 2019 heat wave in Western Europe, , (July) [online] Available from: https://public.wmo.int/en/media/news/july-heatwave-has-multiple-impacts, 2019.

de Villiers, M. P.: Europe extreme heat 22–26 July 2019: was it caused by subsidence or advection?, Weather, 75(8), 228–235, doi:https://doi.org/10.1002/wea.3717, 2020.

Xu, P., Wang, L., Liu, Y., Chen, W. and Huang, P.: The record-breaking heat wave of June 2019 in Central Europe, Atmos. Sci. Lett., 21(4), e964, doi:https://doi.org/10.1002/asl.964, 2020.

The introduction does not provide a clear justification why this work is needed, does not show gaps in current understanding and does not formulate a clear hypothesis.

R: In the revised version of the manuscript we have improved to improve the introduction part in order to make the justification of the paper more clear.

The choice of more than 5 days defining a heatwave is not objectively based (lines 109/110). My suggestion would be to consult the latest literature that deal with more objective measures how heatwaves are defined.

R: Our choice of 5 days was based on the recommended thresholds for the regions surrounding Romania. Moreover, we also followed the recommendations of the Expert Team on Climate Change Detection and Indices (ETCCDI).

Heatwave results reported (lines 149-150, 4 to 5 days) are in disagreement with the definition provided in lines 109/110 that state more than 5 days.

R: In the respective line we speak about an average at country level, meaning that the number is an average over a certain number of grid points. Some grid points might fulfil the heatwave criteria, some not, thus when you average them you will get a number averaged over a large region.

Lines 160/161, there is an overlap of having the year 1970 in both periods. In addition, the period 1970-1985 has a different length compared to others that makes the comparison difficult.

R: Here we compared periods with different characteristic. Is not our choice that the period 1971 –1985 is HW free. We have tried just to comment what was captured by Figure 2.

SPI is not the most appropriate measure for drought. For the area, SPEI is a better index that combines temperature and precipitation.

R: In the revised version of the manuscript we have replaced the SPI with SPEI and the text and the results have been described accordingly.

For a review please consult Raible et al. (2017): Drought indices revisited – improving and testing of drought indices in a simulation of the last two millennia for Europe, Tellus A: Dynamic Meteorology and Oceanography, 69, 1287492.

R: We do not think that this paper is actually relevant for giving a clear suggestion which index is optimal, mainly because it's a modeling study and most of the models have issues in properly represent the potential evapotranspiration which is an essential component in computing SPEI. We think the choice of the drought index should reflect what the authors want to analyze. Out aim was to analyze changes in extreme temperature and extreme precipitation and we have tried to identify the proper indices to do so. A more indicate paper in this respect would be Stagge et al., 2017. Based on the analysis of Stagge et al.(2017) there are no significant difference between SPEI and SPI over our analyzed region (Figure 2c in their paper). Nevertheless, following the suggestions of all the reviewers involved in the review process of our manuscript in the revised version replaced the SPI with SPEI and the text and the results have been described accordingly.

The comparison between EOBs and ROCADA does not provide new evidence, it could be skipped and the analysis could be concentrated on EOBs.

R: We have added also the results of ROCADA dataset mainly because in previous studies we got complains that EOBS might not be suitable to make studies in Romania. But in the revised version of the manuscript have to removed the information and figures regarding the ROCADA dataset.

Sentence on lines 122-123 is not clear.

R: The text has been removed from the revised version of our manuscript.

The manuscript states at various places "statistical significant" changes. However, no information on the underlying statistics to test significance is provided. A few maps show significant areas related to trends, however it is missing how those regions are calculated.

R: This was a misfortune form our part. In the revised version of the manuscript we have added in the Methodology part all the statistical test sand the associated references.

Further, the synoptical maps do not have a field significance information and thus they are difficult to interpret.

R: The anomaly maps for the case studies cannot have a significance field because it is just a snapshot for one event in time, thus we cannot perform any statistical significance.

For the composite maps the significance of the anomalies is actually plotted in the figures (see Figures 13 and 14 for example). Again it was a misfortune from our side that we did not write that clearly in the figure caption.

Further, the maps are not unexpected and the processes that lead to drought or heat extremes are well documented in the literature elsewhere.

R: We really disagree with this comment. The fact that processes that lead to drought/heatwaves are well documented in literature, does not mean that scientist should stop doing this kind of research. Each region has it's own particularities. Yes, of course a heat wave will most probably be driven by a blocking system, but this doesn't mean that we have fixed and fully closed the issue of analyzing drivers of extreme events. Again, it seems to be a very accepted approach for most of the already published studies, but for the current study doesn't seem to be accepted. Moreover, we have added also the stability maps in our manuscript (Figure 13) to further add some new info regarding the relationship between heatwaves and their drivers and we consider this is a new way to study this kind of relationship. Thus, we think this comment does not really reflect the findings and analysis from our manuscript.

The conclusions include information from the introduction and duplicate the results. As such, a lot of information is irrelevant and the last paragraph is not a conclusion from the analysis shown.

R: The has been be modified, improved and adjusted to the new figures which have been produced throughout the review process.

**Reviewer 4**

**General comment**

The authors present a comprehensive analysis regarding the spatial-temporal variability of hot and dry summer in Romania, as well as their combined effect (compound events) that can be considered a novelty for this region. Various characteristics of the heat waves (duration, spatial extent and frequency) and drought index (represented by the standardized precipitation index) have been analysed in terms of their long term trends, decadal variability and driving factor triggering their occurrence. The title *"Hotspots for warm and dry summers in eastern Europe, with a focus on Romania"* is not fit very well with the analysis carried out in this article that is done only for Romania and can not be extended over the entire eastern Europe. Therefore, the title could be changed. Same comment for the title of the sections (3.1, 3.2,....).

R: Thank you for your constructive evaluation of our study. In the revised version of the manuscript we will consider all comments and suggestions and we will improved the manuscript accordingly (see detailed responses below). Moreover, the title of the manuscript and the sub-sections have been modified to properly reflect the analyzed region.

The manuscript represent a substantial contribution to the understanding of large-scale mechanisms controlling the variability of extreme clime events (heatwaves and droughts) in a region with complex topographic features such as Romania. The scientific approaches and the applied methods are valid at international standards. The results are discussed in an appropriate and balanced way, including appropriate references. The tables and figures are appropriate to present the results. The conclusions are clear presented and supported by the obtained results.

Considering all these aspects, the manuscript can be published with some minor corrections presented in my specific comments and technical corrections.

R: Once again we than the reviewer for taking time to review our manuscript and for the kind words.

**Specific comments**

1) Pag 13, L429, "in the eastern part of Europe" should be completed with "focusing on Romania" since the heatwaves have been analyzed only over Romania.

R: The text has been modified as suggested.

2) Pag 13, L431-432: The conclusion "i) the length, spatial extent and frequency of HWs in Romania has increased significantly over the last 70 years, for all summer month" is not entirely true; there is a strong decadal variability with a slight decreasing before '70 years (a correct change point could be find by using the Pettit test) and a real (accelerated) increase after 1990. I) and ii) could be combined. This is also in agreement with the author's comments presented in sections 3.1.

R: Modified as suggested.

3) Pag 13, L438-439, "A significant increase in the frequency of hot extremes has been found at country level, with the most affected regions being in the north-western part and the Dobrogea region (Figure

4)". As I see in Fig. 4, this conclusion is true only on seasonal scale (JJA) and the most affected regions are more extended than those mentioned by the authors. There are differences between the individual months: for example, in July, only a small part (eastern regions and some very small south-western areas) exhibit significant trends. In Table S1 is not clear if the trend is computed for the time series of spatial average over the country. This should be clear mentioned.

R: In the revised version of the manuscript we have describe each month/season separately to make the text and the information clear. In Table S1 we have computed the trend based on the index averaged at country level. We are going to re-write the caption of the table to reflect clear what kind of data are used to compute the trends.

4) Pag 13, L459, "The occurrence of HWs in the eastern part of Europe", " the eastern part of Europe" should be replaced by "Romania", see also my previous comments

R: The text has been modified as suggested.

**Technical corrections**

- Pag3, L86 "to determined" change in "to determine"

- Pag4, L111-112 "values < 1", insert "SPI1 after "1-month"

- Pag 5, L158, change "two centuries" with "two decades".

- Pag 7, L233, replace "Europe" with "Romania".

- Pag 12, L414: "August 2015 is also the month with the longest July heatwave (Figure 2c)" something is not correct here: ."July heatwave (Figure 2c)" should be replaced by "August heatwave (Fig. 2e)"

- Pag 12, L417, "periods (i.e. 1950 – 200": 1950 – 200 should be replaced by 1950-2020.

- Pag 12, L427: I think that "To extended the overview also" should be replaced by "To extend...."

- Pag 13, L429, "in the eastern part of Europe" should be completed with "focusing on Romania" since the heatwaves have been analysed only over Romania.

- Pag 14, L475, "IPP report (IPCC, 2021)", IPP should be replaced by IPCC.

- References: please check all references, there are some duplications in the authors' list: example, Badaluta, C.-A., Persoiu, A., Ionita, M., Nagavciuc, V., Bistricean, P.-I. P.-I., Persoiu, A., Ionita, M., Nagavciuc, V. and Bistricean, P.-I. P.-I.: Stable H and O isotope-based investigation of moisture sources and their role in river and groundwater recharge in the NE Carpathian Mountains, East-Central Europe, Isot. Environ. Heal. Stud., 55(2), 1–18,......

R: All the aforementioned technical corrections have been implemented in the revised version of the manuscript.

[revised manuscript text omitted]

The null hypothesis of no trend is rejected if the p-values is lower than 0.05 (significance level of α =0.05).
* indicates a statistically significant trend the 90% confidence level using the Mann–Kendall test.

*Table S3*. Results of the trend analysis for the monthly Z500 indices (Figure 13 – right column). The trend analysis was conducted based on nonparametric Mann-Kendall test, for two distinct period: 1950 – 2020 and 1990 - 2020.

| | | Z500 Index Trend | P-value |
|---|---|---|---|
| June | 1950 – 2020 | 2.8m/decade | 0.06 |
| | 1990 – 2020 | 12.5m/decade | 0.005* |
| July | 1950 – 2020 | 2.8m/decade | 0.009* |
| | 1990 – 2020 | 7.5m/decade | 0.001* |
| August | 1950 – 2020 | 4.3m/decade | 0.0007* |
| | 1990 – 2020 | 11.5m/decade | 0.006* |

The null hypothesis of no trend is rejected if the p-values is lower than 0.05 (significance level of α =0.05).
* indicates a statistically significant trend the 90% confidence level using the Mann–Kendall test.

[Figure]

***Figure S1.*** June decadal frequency of the number of heat waves (HWs) per decade over the last 70 years: a) 1951 – 1960; b) 1961 – 1970; c)  1971 – 1980; d) 1981 – 1990; e) 1991 – 2000; f) 2001 – 2010 and g) 2011 – 2020. Units: number of HWs/decade.

[Figure]

***Figure S2.*** July decadal frequency of the number of heat waves (HWs) per decade over the last 70 years: a) 1951 – 1960; b) 1961 – 1970; c)  1971 – 1980; d) 1981 – 1990; e) 1991 – 2000; f) 2001 – 2010 and g) 2011 – 2020. Units: number of HWs/decade.

[Figure]

***Figure S3.*** August decadal frequency of the number of heat waves (HWs) per decade over the last 70 years: a) 1951 –
1960; b) 1961 – 1970; c) 1971 – 1980; d) 1981 – 1990; e) 1991 – 2000; f) 2001 – 2010 and g) 2011 – 2020.
Units: number of HWs/decade.

[Figure]

***Figure S4.*** The correlation coefficient between the monthly SPEI and monthly HWDI with different time lags and in phase. The combinations (SPEI and HWDI) which are used in the study to compude the compund hot and dry (CHD) index are highlighted in yellow boxes.

[Figure]

***Figure S5.*** Evolution of the daily maximum temperature (Tx) anomaly over the period
2.07.2012 – 10.07.2012. The anomalies are computed relative to the base period 1971 – 2000.

[Figure]

***Figure S6.*** Evolution of the daily geopotential height at 500mb (shaded colors) and the location of the 2D atmospheric blocking (contour lines and hashed areas) over the period 2.07.2012 – 10.07.2012.

[Figure]

***Figure S7.*** Evolution of the daily maximum temperature (Tx) anomaly over the period
23.08.2015 – 31.08.2015. The anomalies are compute relative to the base period 1971 – 2000.

[Figure]

***Figure S8.*** Evolution of the daily geopotential height at 500mb (shaded colors) and the location of the 2D atmospheric blocking (contour lines and hashed areas) over the period 23.078.2015 – 31.08.2015.

[Figure]

***Figure S9.*** Evolution of the daily maximum temperature (Tx) anomaly over the period
11.06.2019 – 19.06.2019. The anomalies are compute relative to the base period 1971 – 2000.

[Figure]

***Figure S10.*** Evolution of the daily geopotential height at 500mb (shaded colors) and the location of the 2D atmospheric blocking (contour lines and hashed areas) over the period 11.06.2019 – 19.06.2019.

---

## Author Response (AR2)

**Reviewer 1**

It is my second time reviewer the manuscript now entitled" Hotspots for warm and dry summers in Romania". I believe the authors have done a good work in reviewing the manuscript. Most of my comments and the others reviewers' comments have been addressed. However, I still have some concerns that need to be address:

We thank the reviewer for the constructive evaluation of our study and we are glad we were able to address the comments in a satisfactory manner.

1) Do you think analyzing only June, July, and August SPEI1 index and the August SPEI3 index, is enough in lagging the effect of drought in HWDI? My suggesting is to use also SPEI6 (and add also the spring months) since the soil-moisture availability in spring can also influence the occurrence heat waves in early and late summer. I would consider using at least SPEI6 index as done in Russo et al., 2019 which account the soil-moisture availability in the previous seasons.

R: We have tested the risk and correlation maps with different lags and different accumulation periods for the SPEI index (i.e. 1-month, 3-months and 6-months). Since we have produced more than 50 maps just for this kind of analysis, we were unable to actually show the results for each of them. At the end we had to decide and show just the ones which show the highest correlation. We have included the lagged correlation between the HW index and the SPEI for different accumulation periods (for the country based time series) in Figure S4 (now including also SPEI6). Nevertheless, the local hydroclimatology, indicates that the highest correlations are obtained for the in-phase analysis, and similar results have been also found in the Russo et al 2019 paper. Over Romania, it seems that previous spring soil moisture condition are not a necessary ingredient for the occurrence of heatwaves in the upcoming months. There might be some small influence from the spring to summer and we have discussed this information in Section 3.3.

2) The quality of the figures needs to be improved. Figure 2, Figure 5 need to be re-drawn using other software than Excel. In addition, in every spatial map, the latitude and longitude need to be included.

R: Figures 2 and 5 have been produce with OriginLab Pro, which it's a rather complex software and we have saved the figures at a resolution of 1200 DPI. It might be the case that the quality of Figure 2 and 5 is not so good in the manuscript because they have been inserted directly in the document, without any other modifications. We are confident that throughout the typesetting of the paper this issue will be solved. We have added also the latitude and longitude in all the spatial maps.

3) Figures 10 to 12 – Panel f) need to be acknowledge. Please add in the caption that f) refers to water vapor flux divergence. However, I find the water vapor flux divergence field too noisy and a bit hard to extract some information from it. I would delete it from the figures and from the text.

R: The water vapor flux divergence field has been removed from the figures, although some mentions to it has been kept in the text, without showing the figures.

**Reviewer 2**

The paper presents an assessment of the spatio-temporal variability and trends of hot and dry summers, analyzing the physical mechanisms driving the occurrence of hot summers in Romania. For this, the heatwave duration index (HWDI), the Standardized Precipitation and Evapotranspiration index (SPEI) and the compound hot and dry index (CHD) are computed for this region and analyzed. I consider that the manuscript is right for its publication in HNESS.

We thank the reviewer for the constructive evaluation of our study and we are glad we were able to address the comments in a satisfactory manner.

Some technical corrections are listed below:
Page 4, line 128: I think that "was extracted" should be eliminated.
Modified as suggested.

Page 5, line 144: "daily geopotential height at 500mb" should be changed by "Z500".
Modified as suggested.

Page 5, line 144: The reference of Hersbach et al., 2020 should be eliminated here, because it was previously cited.
Modified as suggested.

Page 5, line 145: "perido1950-" should be changed by "period 1950-"
Modified as suggested.

Page 5, line 145: The spatial resolution of the ERA5 data has been already described in line 142.
Modified as suggested.

Page 5, line 151: "where is the latitude…" should be changed with "where $\phi_0$ is the latitude…"
Modified as suggested.

Page 5, line 151: "75°" must be "75°N"
Modified as suggested.

Page 5, line 159: "by computed" must be changed with "by computing".
Modified as suggested.

Page 5, line 161: "AREA" should be "area"? Please, check the need of capital letters, or to define the AREA index previously to its use.
Modified as suggested.

Page 6, line 203: "county" must be "country".
Modified as suggested.

Page 8, line 251: "Figure4c" should be "Figure 4c".
Modified as suggested.

Page 8, line 265: "driest summer" should be "driest summers".
Modified as suggested.

Page 10, line 322: the different Mountains could be pointed in Figure 1. This will help to the location of them.
Modified as suggested.

Page 10, line 349: In Figure 10, the Figure caption describing what is shown in Figure 10f (colour bar and vectors), would be added. Similarly for Figure 11f.
The water vapor figure has been removed from the new version of the manuscript.

Page 15, line 501: "this extreme events" should be "these extreme events".
Modified as suggested.